# Interferon regulatory factor 4 mediates nonenzymatic IRE1 dependency in multiple myeloma cells

Ioanna Oikonomidi[1], Vasumathi Kameswaran[2], Victoria C. Pham[2], Iratxe Zuazo-Gaztelu[1], Lauren M. Gutgesell[1], Scot Marsters[1], Bence Daniel[2], Jennie R. Lill[2], Zora Modrusan[2], Avi Ashkenazi [1]*

1 Department of Research Oncology, Genentech, Inc., South San Francisco, California, United States of America, 2 Department of Proteomic and Genomic Technologies, Genentech, Inc., South San Francisco, California, United States of America

* aa@gene.com

## Abstract

Multiple myeloma (MM) arises through oncogenic transformation of immunoglobulin-secreting plasma cells. MM often co-opts the central endoplasmic reticulum (ER)-stress mitigator, inositol-requiring enzyme 1 (IRE1), to sustain malignant growth. While certain MMs require enzymatic IRE1-dependent activation of the transcription factor XBP1s, others display a nonenzymatic IRE1 dependency that is not yet mechanistically understood. Here we identify interferon regulatory factor 4 (IRF4), which stimulates genes that promote immune-cell proliferation, as a key conduit for IRE1's nonenzymatic control of cell-cycle progression in MM. IRE1 silencing increased inhibitory S114/S270 phosphorylation on IRF4, disrupting IRF4's chromatin-binding and transcriptional activity. IRF4 knockdown recapitulated, whereas IRF4 repletion reversed, the anti-proliferative phenotype of IRE1 silencing. Furthermore, phospho-deficient, but not phospho-mimetic, IRF4 mutants rescued proliferation under IRE1 silencing. Functional studies revealed that IRF4 engages the *E2F1* and *CDC25A* genes and promotes CDK2 activation to drive cell-cycle progression. Our results advance mechanistic understanding of IRE1 and IRF4 in MM.

## Introduction

Genome instability, mutational burden, and metabolic restrictions induce endoplasmic-reticulum (ER) stress in tumor cells [1–5]. In response, ER-resident inositol-requiring enzyme 1 α (herein, IRE1) functions via a conserved cytoplasmic kinase-endoribonuclease (RNase) module [1] to restore homeostasis. The IRE1 kinase mainly acts to regulate IRE1's RNase activity [6–8]. In turn, the IRE1 RNase edits the mRNA encoding X-Box binding protein 1 (XBP1), leading to the generation of "XBP1 spliced" (XBP1s) via non-conventional splicing [1]. XBP1s then acts as a

**Data availability statement:** Proteomics and phospho-proteomics data are available in the MassIVE repository with the dataset identifiers: MSV000095907 (doi:10.25345/C5WH2DS2R) and MSV0000939092 (doi:10.25345/C5XP6VF07). RNAseq and ChIP-seq data from this study can be found in the NCBI GEO repository with accession numbers GSE288674 and GSE288671, respectively. Also, previously published RNA-seq data reanalyzed in this study can be found in GEO with accession number GSE285981. For Fig 5A heatmap related to the ChIP-seq differential binding analysis, underlying analyzed data can be found in "S2 Data" matrices. The FCS files from the FACS experiments can be found in Zenodo (https://doi.org/10.5281/zenodo.14928364). All other numerical data is provided in "S1 Data" file while all uncropped gel/blot images are provided in "S1 Raw Images" file.

**Funding:** The author(s) received no specific funding for this work.

**Competing interests:** I have read the journal's policy and the authors of this manuscript have the following competing interests: All authors were employees of Genentech, Inc. during the performance of this work.

**Abbreviations:** CAN, acetonitrile; CFSE, carboxyfluorescein succinimidyl ester; CID, collision-induced dissociation; DEG, differentially expressed genes; Dox, doxycycline; ER, endoplasmic reticulum; EV, empty vector; FDS, false discovery rate; GO, gene ontology; GPP, global proteome profiling; GSEA, gene-set enrichment analysis; IAA, iodoacetamide; IB, immunoblot; IRE1, inositol-requiring enzyme 1; IRF4, interferon regulatory factor 4; KI, kinase inhibitor; MM, multiple myeloma; ORA, over-representation analysis; PI, propidium iodide; RI, RNase inhibitor; SPS, synchronous precursor selection; STR, short tandem repeat; TFA, trifluoroacetic acid; UPR, unfolded protein response; WT, wildtype; XBP1, X-Box binding protein 1.

potent transcription factor to expand ER capacity and mitigate ER stress. Furthermore, IRE1's RNase cleaves multiple additional mRNAs via a stringent, endomotif-based mechanism called regulated IRE1-dependent decay (RIDD) [9], and through a more promiscuous process dubbed RIDD lacking endomotif (RIDDLE) [7].

Multiple myeloma (MM) is a plasma-cell cancer with malignant characteristics that so far have rendered it clinically incurable. MM cells retain a clonal immunoglobulin (Ig)-secretory phenotype. Hence, they display constitutive ER stress and increased levels of IRE1 and XBP1s [10,11]. Indeed, MM expresses high levels of IRE1, and the IRE1-XBP1 pathway has emerged as a key player in MM pathogenesis [10,12,13]. Experimental IRE1 loss of function attenuates growth of certain MM cell lines *in vitro* and *in vivo* [10,13,14]. Dependency on IRE1 in MM has been attributed primarily to XBP1s because B-lineage-specific transgenic overexpression of XBP1s in mice induces an MM-like phenotype [12]. Moreover, in certain MM models, XBP1 silencing mimics IRE1 disruption [10,13]. However, recent work by our laboratory has identified several MM cell lines that depend on IRE1 but not on XBP1s itself, or more generally, on IRE1's enzymatic activity [15]. How IRE1 mechanistically controls MM growth in such models remains unknown.

The transcriptional regulator interferon response factor 4 (IRF4) orchestrates B-cell activation as well as plasma-cell differentiation and survival [16–20]. During plasma-cell differentiation, IRF4 cooperates with the IRE1-XBP1s pathway, most prominently by inducing transcription of *PRDM1*/BLIMP1 [21,22]. BLIMP1, in turn, de-represses XBP1s [21,23,24], which is essential for Ig secretion [23,25,26]. Previous work has established IRF4 as central to MM pathogenesis, owing to its transcriptional control of gene programs that promote cell-cycle progression and cell survival [17,27]. Furthermore, *IRF4* gene amplification predicts poor prognosis in MM [28]. While some MMs have chromosomal translocations that juxtapose the *IgH* enhancer to the *IRF4* locus, *IRF4* gene alterations are infrequent, altogether accounting for less than a quarter of cases [29–31]. Additional molecular mechanisms that drive IRF4 overexpression in MM include transcriptional self-induction [16] and activation by Myc [17]. Loss of XBP1s indirectly elevates both IRF4 and BLIMP1 [32], indicating transcriptional cross-regulation. Here, we report that IRF4 plays a key role in mediating IRE1's nonenzymatic control of cell-cycle progression and proliferation in MM. Our study advances the current mechanistic and functional understanding of both IRE1 and IRF4 in MM cells.

## Results

### Silencing of IRE1 but not XBP1 downregulates IRF4

We recently discovered an unexpected nonenzymatic dependency on IRE1 in a number of MM cell lines, namely AMO1, KMS27, L363, and JJN3 [15]. Although some of these lines displayed weaker IRE1 and/or XBP1 dependency in published massive-scale genome-wide functional screens (DepMap.Org Portal; [33]), our analysis by inducible shRNA silencing revealed greater dependency on IRE1 as compared to XBP1 under exponential-phase growth *in vitro* and *in vivo* [15]. IRE1

silencing in some of these models caused tumor regression, whereas pharmacologic IRE1 inhibition did not impair tumor growth despite effective blockade of IRE1's enzymatic activity [15]. To identify genes that might mediate nonenzymatic IRE1 dependency in MM, we combined data from DepMap.Org (Chronos model, which assigns cell-fitness-effect scores of 0 or −1, respectively, to non-essential or essential genes) with our previously obtained transcriptomic and proteomic results from IRE1-silenced MM cells [15]. We first determined the "corrected" Chronos gene effect based on unique essentiality in MM *versus* other cancers. We then focused on those genes that were essential in all of the four cell lines displaying nonenzymatic IRE1 dependency, thus identifying 131 specific IRE1 codependencies (S1 Table). We subsequently filtered these 131 genes further based upon the reliance of their mRNA expression upon IRE1 in AMO1 cells (Fig 1A). *IRF4* had the lowest corrected gene-effect score (Fig 1B; S1 Table; [33]), highlighting it as the most crucial among IRE1-regulated codependency genes in MM. RNA-seq data from AMO1 cells [15] further showed that doxycycline (Dox)-induced shRNA silencing of IRE1, but not XBP1, significantly downregulated the *IRF4* mRNA (Fig 1C). Corresponding proteomics data [15] extended this depletion to the IRF4 protein (Fig 1D). Further analysis of the AMO1, KMS27, and L363 cell lines verified the depletion of *IRF4* mRNA and protein upon knockdown of IRE1 but not XBP1 (Fig 1E and 1F and S1A–B Fig). To examine the regulation of IRF4 in a model that does require IRE1's enzymatic activity, we investigated the well-characterized XBP1-dependent MM cell line, KMS11 [10]. Of note, KMS11 also displays stronger MM-specific dependency on *IRF4 versus* other genes (S1C Fig). Either IRE1 or XBP1 knockdown in KMS11 cells led to downregulation of IRF4 (S1D Fig), consistent with enzymatic IRE1 regulation of IRF4 via XBP1s. For further comparison, we also investigated IRF4 regulation in the IRE1-independent MM cell line, H929 (S1E–F Fig), known to be strongly dependent on IRF4 [17,27]. IRE1 depletion in H929 cells did not affect IRF4 abundance (S1F Fig). Additionally, we examined the IRE1-independent T-cell lymphoma cell line, SR786 (S1G Fig), which also showed little change in IRF4 levels upon IRE1 knockdown (S1H Fig). These data suggest a more prominent IRE1-mediated regulation of IRF4 in IRE1-dependent *versus* IRE1-independent MM cell lines. Furthermore, IRE1 can regulate IRF4 either nonenzymatically, through an as yet unidentified scaffolding modality, or enzymatically, via XBP1s (S1I Fig). To verify that the regulation of IRF4 by IRE1 in AMO1 and KMS27 cells was indeed nonenzymatic, we used a small-molecule IRE1 RNase inhibitor, which blocked both *XBP1* splicing and RIDD (S1J–K Fig). Enzymatic IRE1 inhibition did not decrease IRF4 levels; rather, in AMO1 cells it increased IRF4 abundance (Fig 1G–1J), in keeping with previous evidence that IRF4 can be targeted by RIDD [34]. Together, these results identify IRF4 as a potential mediator of nonenzymatic IRE1 dependency in MM cells and demonstrate that IRE1 can nonenzymatically regulate IRF4 expression independently of XBP1s and RIDD.

## IRE1 silencing attenuates IRF4 activity

The depletion of both IRF4 mRNA and protein upon IRE1 knockdown suggested that IRE1 may control IRF4 abundance at the mRNA level. However, an actinomycin-D chase indicated that IRE1 silencing only slightly accelerated *IRF4* mRNA degradation (S2A Fig), suggesting the existence of additional regulatory mechanisms. Analysis of nascent mRNA through nuclear run-on revealed that IRE1 silencing significantly reduced *de novo IRF4* transcription (Fig 2A), affecting this particular feature more substantially than mRNA turnover (S2A Fig). We therefore interrogated several transcription factors known to regulate IRF4 [35] either negatively (i.e., MITF), or positively (i.e., Myc, Ikaros/*IKZF1*, and Aiolos/*IKZF3*). However, IRE1 knockdown downregulated *MITF* mRNA and protein (Fig 2B), rendering MITF an unlikely driver of the reduction in IRF4. Furthermore, IRE1 silencing did not downregulate *IKZF1* or *IKZF3* (Fig 2C), and although it did deplete the Aiolos protein (Fig 2D), this occurred later than IRF4 downregulation. We could not detect the Ikaros protein in these cells. Thus, these latter factors are also unlikely to mediate the downregulation of IRF4 conferred by IRE1 depletion. Moreover, although IRE1 silencing did decrease *Myc* mRNA and protein abundance (Fig 2D and S2B Fig), this again occurred later than the onset of IRF4 downregulation, suggesting that Myc depletion does not drive IRF4 loss. To further examine these factors independently of their mRNA levels, we assessed their chromatin-binding activity through subcellular fractionation (S2C Fig). Neither MITF nor Aiolos displayed altered chromatin binding upon IRE1 knockdown, whereas Myc showed

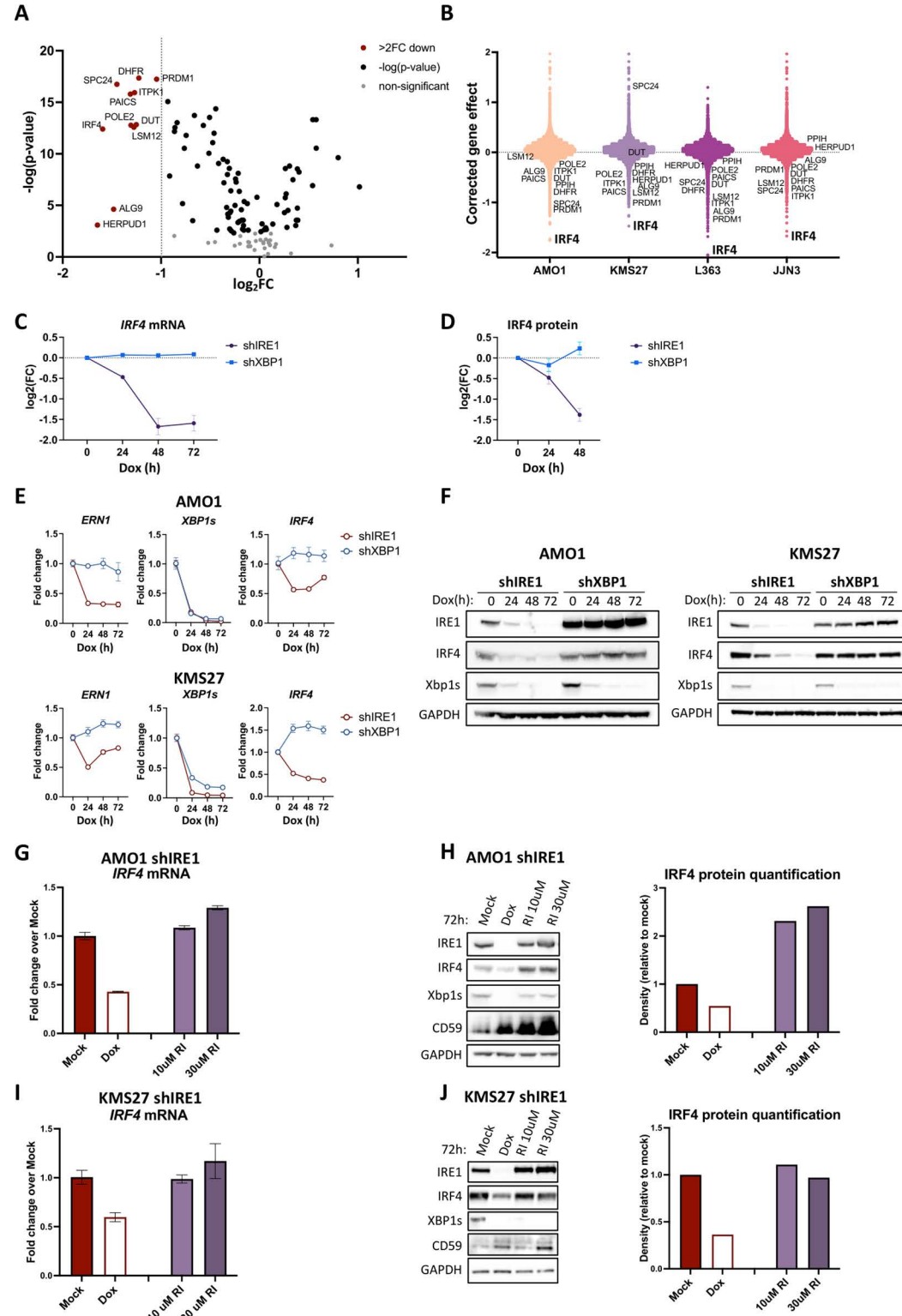

**Fig 1. Silencing of IRE1 but not XBP1 downregulates IRF4.** (A) Effect of IRE1 silencing on MM-specific dependencies. Chronos gene-effect values derived from Depmap were "corrected" as described in *Methods* to reveal MM-specific dependencies. Subsequently, nonenzymatic-IRE1 co-dependencies were obtained by filtering for those MM-specific essential genes that were essential (*corrected* score ≤ −0.25) in all nonenzymatic-IRE1 dependent cell

lines (S1 Table, 131 genes). Transcriptomic analysis of AMO1 shIRE1 Cl. 1 cells (Dox-inducible shIRE1) treated with 0.2 µg/mL Dox for 72 h [15] filtered for the 131 MM-specific essential genes that are also nonenzymatic-IRE1 co-dependencies (S1 Table). Differentially expressed MM-specific essential genes exceeding a 2-fold decrease are colored in red. (B) *Corrected* gene-effect scores of IRE1-dependent genes: Chronos gene-effect values derived from Depmap were "corrected" as above to reveal MM-specific dependencies. Annotated are genes decreased 2-fold or more by IRE1 silencing in AMO1 cells (red in Fig 1A). (C) Effect of IRE1 silencing on *IRF4* mRNA as indicated by RNA-seq. Normalized IRF4 transcripts throughout a time course of IRE1 or XBP1 silencing. Data was obtained from the transcriptomic study published previously [15], represented as mean ± SEM of Log base 2 fold-change (FC). (D) Effect of IRE1 silencing on IRF4 protein levels. Normalized peptide counts of IRF4 in a time-course of IRE1 vs. XBP1 silencing in AMO1 cells (AMO1 shIRE1 Cl.1 vs. AMO1 shXBP1 Cl.1). Data was obtained from the proteomic study published previously [15], represented as mean ± SEM of Log base 2 FC. (E) Effect of IRE1 or XBP1 silencing on *IRF4* mRNA abundance in AMO1 and KMS27 cells. AMO1 shIRE1 Cl.1 vs. AMO1 shXBP1 Cl.1 cells or KMS27 shIRE1 Cl.9 vs. KMS27 shXBP1 Cl. 13 cells were treated with Dox (0.2 µg/mL) for the indicated time and analyzed by RT-qPCR for levels of *ERN1* (IRE1 transcript), *IRF4*, and *XBP1u*. Data represented as mean ± SEM. (F) Effect of IRE1 or XBP1 silencing on IRF4 protein abundance in AMO1 and KMS27 cells. AMO1 shIRE1 Cl.1 vs. AMO1 shXBP1 Cl.1 cells or KMS27 shIRE1 vs. KMS27 shXBP1 cells in a time-course of Dox (0.2 µg/mL) treatment for up to 72 h were analyzed by immunoblot (IB) for IRF4 protein levels. (G) Effect of IRE1 RNase inhibition on *IRF4* mRNA abundance in AMO1 cells. AMO1 shIRE1 Cl.1 cells were treated with Dox or a specific IRE1-RNase inhibitor (RI: 4µ8C, 10 or 30 µM) for 72 h and analyzed by RT-qPCR for *IRF4* mRNA. Data represented as mean ± SEM. (H) Effect of IRE1 RNase inhibition on IRF4 protein abundance in AMO1 cells. Samples from (G) were analyzed by IB for IRF4 protein. XBP1s and CD59 (RIDD target) were used as controls for effective IRE1 inhibition. *Right:* IRF4 protein quantification of the presented IB by densitometry. IRF4 values were normalized to GAPDH and then expressed relative to IRF4 levels in Mock samples. (I) Effect of IRE1 RNase inhibition on *IRF4* mRNA abundance in KMS27 cells. KMS27 shIRE1 cells were treated with Dox or a specific IRE1-RNase inhibitor (RI: 4µ8C, 10 or 30 µM) for 72 h and analyzed by RT-qPCR for *IRF4* mRNA. Data represented as mean ± SEM. (J) Effect of IRE1 RNase inhibition on IRF4 protein abundance in KMS27 cells. Samples from (I) were analyzed by IB for IRF4 protein. XBP1s and CD59 (RIDD target) were used as controls for effective IRE1 inhibition. *Right:* IRF4 protein quantification of the presented IB by densitometry. IRF4 values were normalized to GAPDH and then expressed relative to IRF4 levels in Mock samples. Data underlying this figure can be found in S1 Data, S1 Raw images, GEO archive (GSE285981) and MassIVE repository (MSV000093902, https://doi.org/10.25345/C5XP6VF07).

decreased activity (S2C Fig). However, despite the evident Myc disruption by IRE1 silencing, shRNA or siRNA-based Myc knockdown did not reduce *IRF4* mRNA or protein expression (Fig 2E and S2D Fig), excluding Myc as a key mediator of IRF4 downregulation upon IRE1 silencing. Together, these results suggest that IRE1 can regulate IRF4 independently of MITF, Ikaros, Aiolos, and Myc (as well as XBP1s).

As previously reported [16,36] and herein confirmed (see below), IRF4 acts as an auto-stimulatory transcription factor. We therefore reasoned that IRE1 may control *IRF4* transcriptionally by regulating IRF4's self-sustaining activity. Supporting this possibility, subcellular fractionation showed that IRE1 silencing markedly decreased the amount of IRF4 bound to chromatin (Fig 2F). Moreover, phospho-proteomic *versus* global proteomic analysis revealed that shRNA-based IRE1 silencing, but not pharmacologic IRE1 inhibition, significantly increased specific IRF4 phosphorylation on Ser114 and Ser270 (Fig 2G), indicating that IRE1 selectively suppresses phosphorylation at these sites. To examine the functional importance of these newly identified sites, we generated IRF4 mutants in which both IRE1-modified serine residues were replaced either with alanine residues (S114A_S270A; phospho-deficient) or with aspartic acid residues (S114D_S270D; phospho-mimetic). Upon ectopic expression at comparable levels in AMO1 cells (S2E Fig), the phospho-deficient mutant displayed a 4-fold increase in chromatin-binding capacity as compared to wildtype (WT) IRF4, while the phospho-mimetic mutant—reflecting the gain of IRF4 phosphorylation upon IRE1 depletion—showed half as much chromatin binding as did WT IRF4 (Fig 2H and S2F Fig), recapitulating IRF4's behavior during IRE1 silencing. Thus, phosphorylation on S114 and/or S270 upon IRE1 silencing significantly suppresses IRF4 activity. Taken together, these results suggest that IRE1 promotes *IRF4* transcription by suppressing inhibitory IRF4 phosphorylation to increase IRF4 chromatin binding and transcriptional self-induction.

## IRF4 silencing recapitulates the anti-proliferative phenotype of IRE1 knockdown

To further validate DepMap's evidence of IRF4's essentiality in the MM models used herein, we generated AMO1 and KMS27 lines expressing Dox-inducible shRNA against *IRF4* (shIRF4) and confirmed efficient IRF4 depletion upon Dox treatment (S3A–B Fig). Consistent with DepMap data, IRF4 knockdown substantially inhibited spheroid growth of these cells, similar to IRE1 knockdown (Fig 3A and S3B Fig).

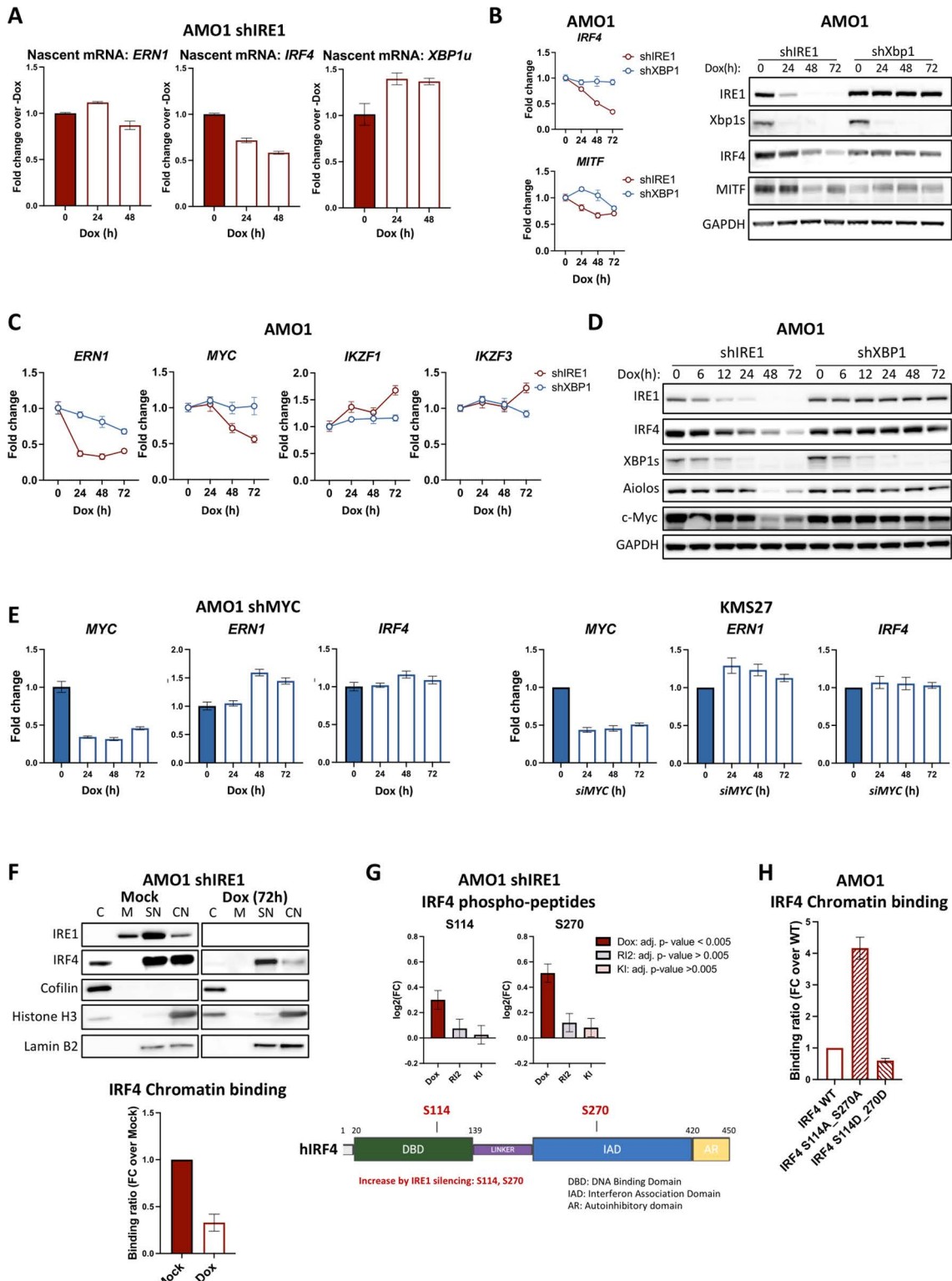

**Fig 2. IRE1 silencing attenuates IRF4 activity.** (A) Analysis of *IRF4* mRNA transcription by nuclear run-on. AMO1 shIRE1 Cl.1 cells were treated with 0.2 μg/mL Dox for the indicated time. Nascent RNA was labeled with a uridine analog, captured, and analyzed for *ERN1*, *IRF4,* and *XBP1*. Representative experiment out of three biological replicates. Data represented as mean ± SEM. (B) Effect of IRE1 and XBP1 silencing on MITF. AMO1 shIRE1

Cl.1 vs. shXBP1 Cl.1 cells were treated in a time-course of Dox (0.2 µg/mL) for up to 72 h and were analyzed by RT-qPCR for *MITF* and *IRF4* mRNA (left), and by IB for MITF protein levels (right). Data represented as mean ± SEM. (C) Effect of IRE1 and XBP1 silencing on Myc and Ikaros/Aiolos at the mRNA level. AMO1 shIRE1 Cl.1 and AMO1 shXBP1 Cl.1 cells were treated in the absence or presence of Dox (0.2 µg/mL) for the indicated time. Samples were analyzed by RT-qPCR for *MYC*, *IKZF1* (Ikaros transcript), and *IKZF3* (Aiolos transcript). Data represented as mean ± SEM. (D) Effect of IRE1 and XBP1 silencing on Myc protein levels. AMO1 shIRE1 Cl.1 and AMO1 shXBP1 Cl.1 cells were treated in the absence or presence of Dox (0.2 µg/mL) for the indicated time and were analyzed by IB for Myc and Aiolos. Ikaros could not be detected. (E) Effect of MYC silencing on IRF4 in AMO1 and KMS27 cells. AMO1 cells were stably transfected with plasmids encoding Dox-inducible shRNA against Myc. AMO1 shMYC Cl. 7 cells (left) were treated with 0.2 µg/mL Dox for the indicated time. KMS27 cells (right) were nucleofected with or without siRNA against *MYC* for the indicated time. Cells were harvested and analyzed by RT-qPCR for *MYC*, *ERN1*, and *IRF4* mRNA. Data represented as mean ± SEM. (F) Effect of IRE1 silencing on IRF4 chromatin-binding. AMO1 shIRE1 Cl.1 cells were cultured in the absence or presence of 0.2 µg/mL Dox for 72 h. Cells were sequentially lysed into 4 subcellular fractions: C—cytoplasmic, M—Membrane, SN—Soluble Nuclear, CN—Chromatin-bound Nuclear. Nuclear fractions were analyzed by IB for IRE1 and IRF4 while Cofilin, Histone H3, and Lamin B2 served as fractionation controls. *Bottom:* The ratio of chromatin-bound over soluble nuclear IRF4 was determined by densitometry in 3 independent biological replicates and is depicted relative to mock-treated cells. (G) Nonenzymatic-IRE1-dependent phosphorylation sites on IRF4. AMO1 shIRE1 Cl.1 cells were treated in the absence or presence of 0.2 µg/mL Dox, 1 µM RI2 (a specific IRE1 RNase inhibitor), or 1 µM KI (a specific IRE1 kinase inhibitor) for 24 h and subjected to phosphoproteomic and global proteomics analyses. The phosphorylation data were normalized against the global proteomics data to control for changes in total protein levels. Four IRF4-phosphopeptides were identified, containing Ser114, Ser270, Ser241, Ser443, and Ser448. Only Ser114 and Ser270 sites were altered in a statistically significant fashion by IRE1 silencing. Top: Fold-change ($log_2$FC) intensity of IRE1-dependent IRF4 phosphopeptides. Data represented as mean ± SE. Student *t* test: *p*-values were adjusted with the Benjamini–Hochberg method. Bottom: IRF4 protein schematic annotating IRE1-dependent phosphorylations. (H) Effect of substitution of S114 and S270 by alanine (phospho-deficient) or aspartic acid (phospho-mimetic) on IRF4's chromatin-binding activity. AMO1 cells ectopically expressing Dox-inducible IRF4 WT, S114A_S270A, or S114D_S270D were incubated with 0.2 µg/mL Dox for 72 h, then fractionated and analyzed by IB (S2F Fig) as in Fig 2F. The ratio of chromatin-bound over soluble nuclear IRF4 was determined by densitometry in three independent biological replicates and is depicted relative to the WT IRF4 chromatin binding ratio. Data represented as mean ± SEM. Data underlying this figure can be found in S1 Data, S1 Raw images, and MassIVE repository (MSV000095907, https://doi.org/10.25345/C5WH2DS2R).

Earlier studies indicated that IRF4 disruption can induce cell-cycle arrest as well as cell death [38,39]. In keeping, depletion of either IRE1 or IRF4 not only inhibited proliferation (Fig 3A and S3B Fig), but also induced apoptosis—evident by increased cleavage of the caspase substrates PARP1 and Lamin A/C, as well as a robust increase in markers of DNA-damage-induced apoptosis, namely, histone H3, H2AX, and γH2AX (S3C Fig). Caspase Glo assays (S3D Fig) and Annexin V/PI staining (S3E Fig) further confirmed the induction of apoptosis. We previously found that although IRE1 silencing induces both cell-cycle arrest and apoptosis, growth deficiency upon IRE1 knockdown stems from loss of proliferation rather than increased cell death [15]. Notably, the pan-caspase inhibitor, Q-VD-Oph (Q-VD), applied at a concentration that effectively prevented multiple apoptotic features (S3F–G Fig), failed to rescue growth upon silencing of either IRE1 or IRF4 (S3H–I Fig). Thus, similar to IRE1 silencing, IRF4 knockdown inhibits proliferation independently of apoptosis.

To further examine proliferation, we quantified cell divisions by staining AMO1 cells with carboxyfluorescein succinimidyl ester (CFSE). We used etoposide as an anti-proliferative control to confirm the lack of CFSE dilution in the absence of cell division (Fig 3B and S3J–K Fig). Both IRE1- and IRF4-silenced cells underwent fewer divisions than did their non-silenced counterparts, whereas XBP1-silenced cells showed little difference *versus* controls (Fig 3B and S3J–K Fig). Additional staining to monitor cell death indicated that those cells that failed to divide proceeded to die (S3K Fig), suggesting earlier proliferative disruption than apoptotic induction in both backgrounds. Bromo-deoxy-uridine (BrdU) incorporation further showed that knockdown of IRE1 or IRF4 progressively attenuated DNA replication with similar kinetics (Fig 3C). To characterize specific cell-cycle phases, we tracked DNA content by cell staining with propidium iodide (PI). Flow cytometry showed that knockdown of either IRE1 or IRF4 in an asynchronous cell population led to G1-phase accumulation at the expense of S-phase by 16 h post-Dox addition (Fig 3D), suggesting rapid G1 arrest. KMS27 cells, which have a longer doubling time, nevertheless displayed G1 arrest by 24 h of IRE1 or IRF4 knockdown (S3L Fig). To examine the extent of Myc's contribution to these changes, we compared IRE1-, IRF4-, or Myc-deficient AMO1 cells for proliferation and cycling capacity (S3M Fig). Remarkably, IRE1 and IRF4 silencing led to a more substantial proliferative inhibition and G1 arrest than did Myc knockdown (S3M Fig), suggesting a less prominent role for Myc as compared to IRE1 and IRF4.

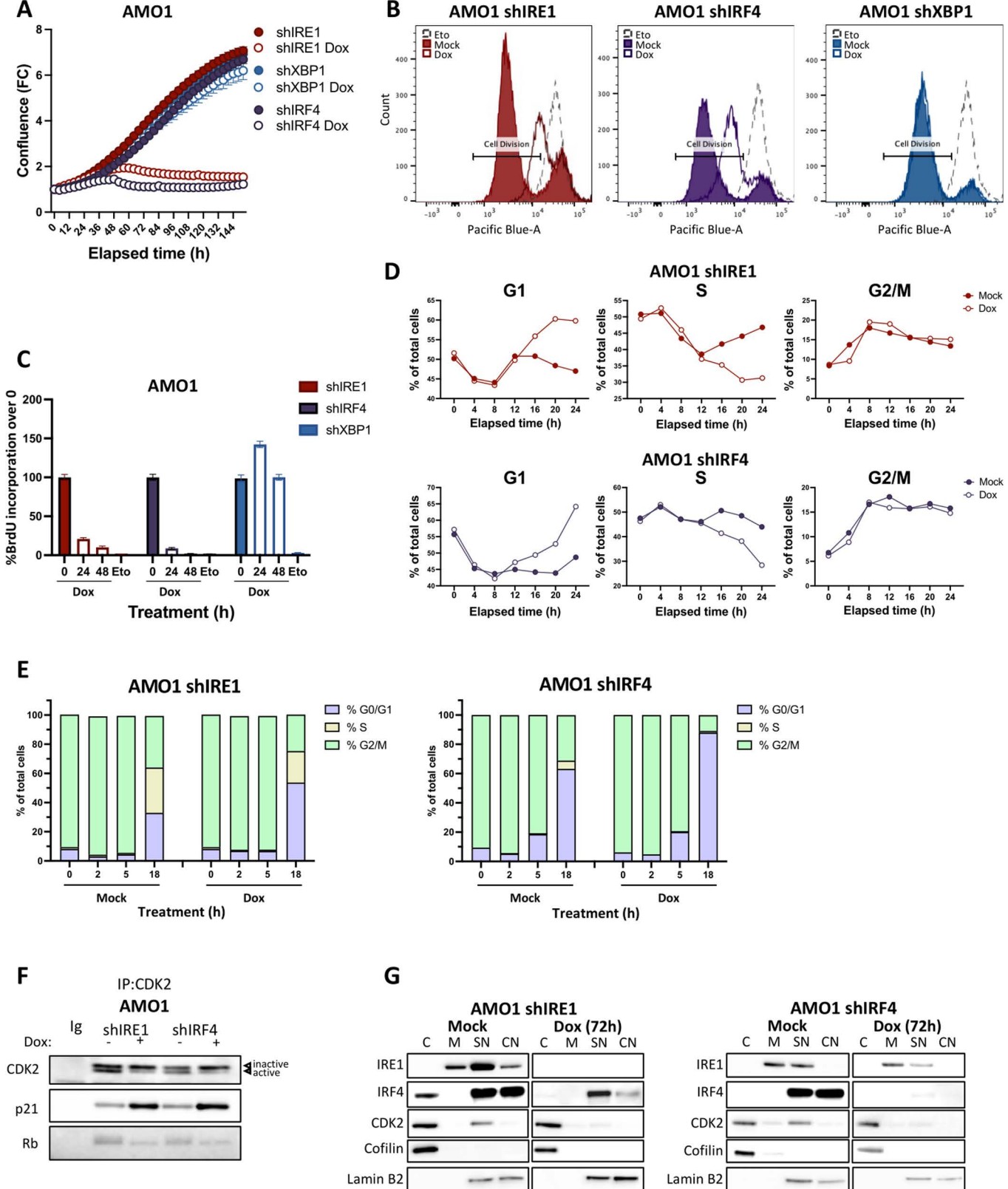

**Fig 3. IRF4 silencing recapitulates the anti-proliferative effect of IRE1 knockdown.** (A) Effect of IRF4, IRE1, or XBP1 silencing on *in vitro* spheroid growth of AMO1. Cells were stably transfected with plasmids encoding Dox-inducible shRNAs against either IRF4 (purple) or non-targeting control

(blue). Growth of these cells in the absence (closed symbols) or presence (open symbols) of Dox (0.2 μg/mL) was compared to that of cells expressing shRNAs against IRE1 or XBP1. Spheroid growth, depicted as FC confluence, was monitored by time-lapse microscopy in an IncuCyte instrument and values represent mean ± SEM. (B) Effect of IRF4, IRE1, or XBP1 silencing on number of cell divisions. AMO1 shIRE1 Cl.1, shIRF4 Cl.1, or shXBP1 Cl.1 cells were stained with CFSE-type dye and incubated in the absence (filled curves) or presence (open curves) of Dox (0.2 μg/mL) and analyzed by flow cytometry. Etoposide (Eto, 25 μM, dashed line) was used as a non-proliferative control. Representative experiment out of 3 independent replicates. (C) Effect of IRF4, IRE1, or XBP1 silencing on DNA replication. AMO1 shIRE1 Cl.1, shIRF4 Cl.1, or shXBP1 Cl.1 cells were pulsed with BrdU (10 μM) and incubated in the absence (filled bars) or presence (open bars) of Dox (0.2 μg/mL) and analyzed by flow cytometry. Etoposide (Eto, 25 μM, dotted line) was used as a non-proliferative control. Data represented as mean ±SEM. (D) Effect of IRE1 or IRF4 silencing on cell cycle progression. AMO1 shIRE1 Cl.1 or shIRF4 Cl.1 cells were incubated in the absence (filled symbols) or presence (open symbols) of Dox (0.2 μg/mL) for the indicated timepoints, EtOH-fixated and PI stained before analyzed by flow cytometry. The indicated cell cycle phases were determined according to univariate (DNA content) modeling. Representative experiment out of at least 3 independent replicates. (E) Effect of IRE1 or IRF4 silencing on the rate of G2/M progression. AMO1 shIRE1 Cl.1 or shIRF4 Cl.1 cells were pre-incubated with 9 μM RO-3306 CDK1 inhibitor (synchronization to G2/M phase) in the absence or presence of Dox (0.2 μg/mL). Cells in G2/M phase were collected and their cell cycle progression during indicated time points post-sorting was analyzed by flow cytometry as before. The indicated cell cycle phases were determined according to DNA content and EdU incorporation to accurately decipher S phase. (F) Effect of IRE1 or IRF4 silencing on CDK2 activation. AMO1 shIRE1 Cl.1 or shIRF4 Cl.1 cells were incubated in the absence or presence of Dox (0.2 μg/mL) for 24 h. CDK2 was purified by immunoprecipitation. The top band is inactive CDK2 and the bottom band is the active form [37]. Additionally, binding of the CDK2 substrate, Rb, is reduced by IRE1 or IRF4 silencing while binding of p21, the CDK inhibitor, is increased. Ig represents an isotype control for Ig detection. (G) Effect of IRE1 or IRF4 silencing on subcellular abundance of CDK2. Samples from Fig 2F and samples from AMO1 shIRF4 Cl.1 cells were analyzed by IB for CDK2 protein. Subcellular fractions: C—cytoplasmic, M—Membrane, SN—Soluble Nuclear, CN—Chromatin-bound Nuclear. Nuclear fractions were analyzed by IB for IRE1 and IRF4 while Cofilin, Histone H3, and Lamin B2 served as fractionation internal controls. The blots for IRE1, IRF4, Cofilin, and Lamin B2 from Fig 2G are shown here again for direct comparison. Data underlying this figure can be found in S1 Data and S1 Raw images.

To better detect potential effects on G2/M, we followed a cohort of G2-synchronized AMO1 cells transitioning first into G1 and then into S-phase (Fig 3E). Not only did these cells undergo G1 arrest, but also their G1 proportion further increased due to faster transition from G2 to the G1 phase of the next cycle (Fig 3E). Thus, the proliferative disruption conferred by IRE1 or IRF4 silencing may be due to defects in both early and late phases of the cell cycle. Substantiating this possibility, already at 24 h after Dox addition, IRE1 or IRF4 silencing inactivated CDK2—a kinase that temporally coordinates both G1/S and G2/M progression and mitosis [40–44] (Fig 3F and S3N–O Fig). CDK2 inactivation was evident by an altered SDS-PAGE migration profile of the CDK2 protein [37] (Fig 3F); as well as by increased CDK2 binding to p21 and decreased CDK2 binding to Rb (Fig 3F). Furthermore, either IRE1 or IRF4 knockdown attenuated stimulatory CDK2 phosphorylation on Thr160 (S3N–O Fig), and IRF4 knockdown augmented inhibitory CDK2 phosphorylation on Thr14 (S3O Fig). One factor that may contribute to the altered CDK2 phosphorylation is the phosphatase CDC25A, which was significantly downregulated upon knockdown of IRF4 (S3P Fig). In keeping with the known nuclear localization of CDK2 in proliferating cells [45], we detected CDK2 in the soluble nuclear fraction of control cells; strikingly, silencing of either IRE1 or IRF4 eliminated CDK2 from this compartment (Fig 3G), providing further evidence of CDK2 disruption. Taken together, these results show that knockdown of either IRE1 or IRF4 leads to a similar anti-proliferative phenotype in IRE1-dependent MM cells, characterized by cell-cycle arrest in conjunction with CDK2 inactivation.

## IRE1 and IRF4 regulate a highly overlapping set of cell cycle genes

Our previous transcriptomic analysis of AMO1 cells revealed marked changes in the expression of multiple cell-cycle genes by 24 h of IRE1 silencing [15]. To assess if IRF4 exerts similar controls, we performed bulk RNA sequencing (RNA-seq) of AMO1 shIRF4 cells treated with Dox for 0, 8, 16, and 24 h and KMS27 shIRF4 cells treated with Dox for 0, 20, and 30 h (to accommodate the slower KMS27 growth rate). For comparison, we used cells expressing non-targeted control shRNA. As expected, IRF4 silencing depleted mRNAs encoding IRF4 itself and other previously characterized IRF4 transcriptional targets, such as MYC [17,46], PRDM1 [16,17,20,22], and XBP1 [20,22] (S4A Fig). For consistency, we re-analyzed the published IRE1 RNA-seq alongside the new IRF4 data obtained here. For both IRE1 and IRF4 knockdown, gene-set enrichment analysis (GSEA) revealed a strong and significant decrease in the Hallmark pathways "G2/M Checkpoint", "E2F Targets", "Myc targets" and "Unfolded Protein Response" (Fig 4A and S4B Fig), corroborating the observed

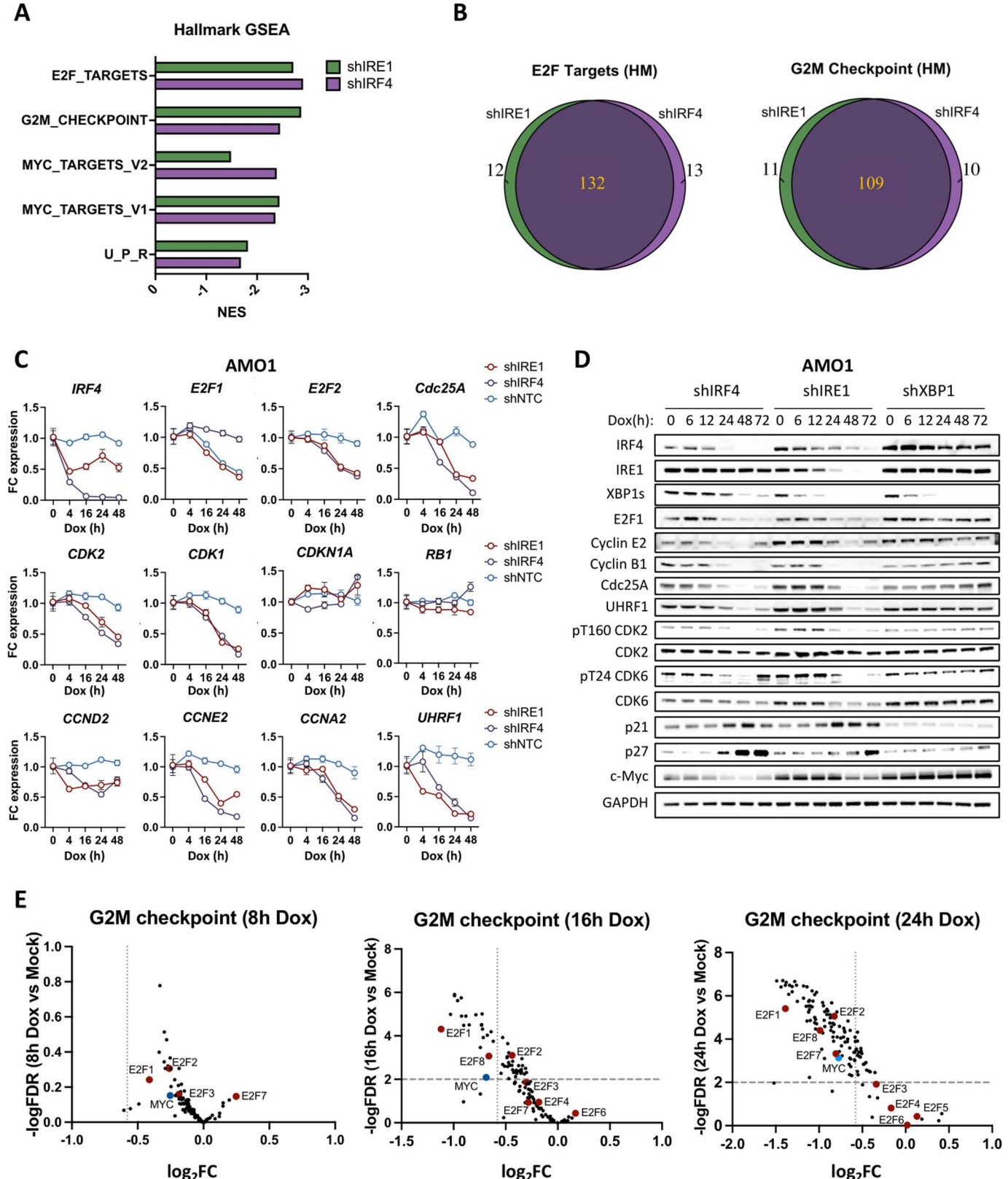

**Fig 4. IRE1 and IRF4 regulate a highly overlapping set of cell cycle genes.** (A) Comparative GSEA of IRE1 and IRF4. AMO1 shIRF4 Cl.1 or shNTC cells were incubated for 0, 8, 16, and 24 h with Dox (0.2 μg/mL) in biological triplicates and subjected to bulk RNA sequencing. DEGs in the shNTC

samples were removed from the analysis as non-specific hits. Shown are the significant (FDR < 0.02) results stemming from Gene Set Enrichment Analyses (GSEA) using Hallmark libraries for both IRF4 and IRE1 silencing of 24 h (shIRE1 data obtained previously [15] but re-analyzed for consistency). For GSEA analysis parameters used, see S1 Data. (B) Overlap of Leading Edge Genes from GSEA in IRE1 and IRF4 knockdowns. After GSEA in Fig 4A, Leading Edge Genes (S2 Table), representing those contributing most significantly to the enrichment of the given gene sets, were extracted for both genetic backgrounds. Shown: Venn diagrams illustrating the intersection between the Leading Edge Genes for "E2F-targets" and "G2/M Checkpoint" between IRE1 knockdown (shIRE1) and IRF4 knockdown (shIRF4) genetic backgrounds. (C) Validation of cell cycle-related gene downregulation by RT-qPCR. AMO1 shIRE1 Cl.1, shIRF4 Cl.1, or shNTC cells were treated with Dox (0.2 μg/mL) for the indicated times and samples were analyzed by RT-qPCR for *E2F*s, *CDK*s, *Cdc25A, CDKN1A* (p21 transcript), *CCN*s (Cyclin transcripts), and *RB1*. Data represented as mean ± SEM. (D) Validation of cell cycle-related gene downregulation by IB. AMO1 shIRE1 Cl.1, shIRF4 Cl.1, or shXBP1 Cl.1 cells were treated with Dox (0.2 μg/mL) for the indicated time and samples were analyzed by IB for cell cycle regulatory proteins. (E) Volcano plot analysis of "G2/M checkpoints" genes in a time-course of IRF4 silencing (8 h: before the onset of cell cycle defects; 16 h: at the onset of cell cycle defects; 24 h: cell cycle arrest established). *E2F* transcripts are indicated in red whereas *Myc* is in blue. Data underlying this figure can be found in S1 Data, S1 Raw images, and GEO archive (GSE288674).

cell-cycle disruption (Fig 3). Over-representation analysis (ORA) revealed additional relevant gene-ontology (GO) terms, including "Mitotic cell process" and "DNA repair" (S4C and S4F Fig). To further gauge the overlap between differentially expressed genes (DEGs) dependent on IRE1 or IRF4, we performed a GSEA Leading-Edge analysis (S2 Table). DEGs in the top cell-cycle-related molecular signatures (E2F Targets and G2M Checkpoint) overlapped by approximately 84%, with 91% of the IRF4 DEGs being shared by IRE1 (Fig 4B). Wiki Pathway analysis of "G1/S Cell Cycle Control" indicated a similar overlap (S4D Fig), consistent with a potential mechanistic linkage between IRE1 and IRF4 in cell-cycle control. Furthermore, 95% of the genes underlying the "DNA repair" term showed overlapping dependency on IRE1 and IRF4 (S4D Fig). In contrast, while both IRE1 and IRF4 knockdown downregulated the "Unfolded Protein Response" (UPR) gene set (Fig 4A) and led to a similar decline in the UPR mediators PERK and ATF6 (S4E Fig), the two backgrounds showed only 45% overlap (S4D Fig), suggesting different entry points into the UPR pathway.

In keeping with the GSEA results, multiple cell-cycle genes, including several E2Fs, also showed perturbation at the protein level (Fig 4D and S3P Fig). Of note, MYC protein was more refractory to changes under either of the two backgrounds (Fig 4D), suggesting a plausible explanation for why targets of MYC were not prominent among enriched gene sets (Fig 4A). The loss in E2F function, evident by downregulation of multiple E2F targets and concomitant modulation of several cell-cycle regulators (Fig 4C and 4D), could decrease CDK2 activity through a number of mechanisms, including de-repression of the CDK inhibitor *CDKN1A*/p21, and downregulation of Cyclins [47] and Cdc25A [48] (Fig 4C and 4D and S3P Fig).

### IRF4 directly controls E2F1 gene transcription

*E2F1* was one of the earliest G2/M-checkpoint transcripts that IRF4 silencing decreased, and its downregulation preceded and persisted beyond the onset of cell-cycle arrest (Fig 4E). Additionally, in IRF4-depleted KMS27 cells, the most down-regulated pathway was "E2F Targets" (S4G Fig). Notably, IRF4 knockdown displayed less DEG overlap between AMO1 and KMS27 cells (S4G Fig), likely reflecting the distinct mutational background of these two lines. Nevertheless, in both instances, IRF4 silencing specifically depleted *E2F1* and some of its downstream targets, such as *Cyclins* and *Cdc25A* (Fig 4C–D).

*E2F1* and *E2F2* [27], as well as *E2F5* [17], have been identified as direct IRF4 target genes. To determine how IRF4 controls *E2F1* in IRE1-dependent cells, we performed a chromatin immunoprecipitation and sequencing (ChIP-seq) analysis. Using this approach, we compared IRF4 with the active, Serine-2-phosphorylated form of RNA-polymerase II (RNAPIIpS2, engaged in transcriptional elongation) (Fig 5A). IRF4 silencing in AMO1 cells decreased both IRF4 and RNAPIIpS2 binding at 868 sequence-specific sites (Fig 5A), indicating IRF4-dependent regulation of RNAPII binding and activity at these genomic locations. *De novo* transcription-factor motif discovery using Homer [49] indicated the enrichment of IRF, E2A, ETS-IRF-composite, RUNX, and E2F binding sites in these differentially bound regions (Fig 5A). At least four of these correspond to the *PRDM1* locus (S5A Fig)—a gene that we found to be regulated by IRF4 (S4A Fig). By

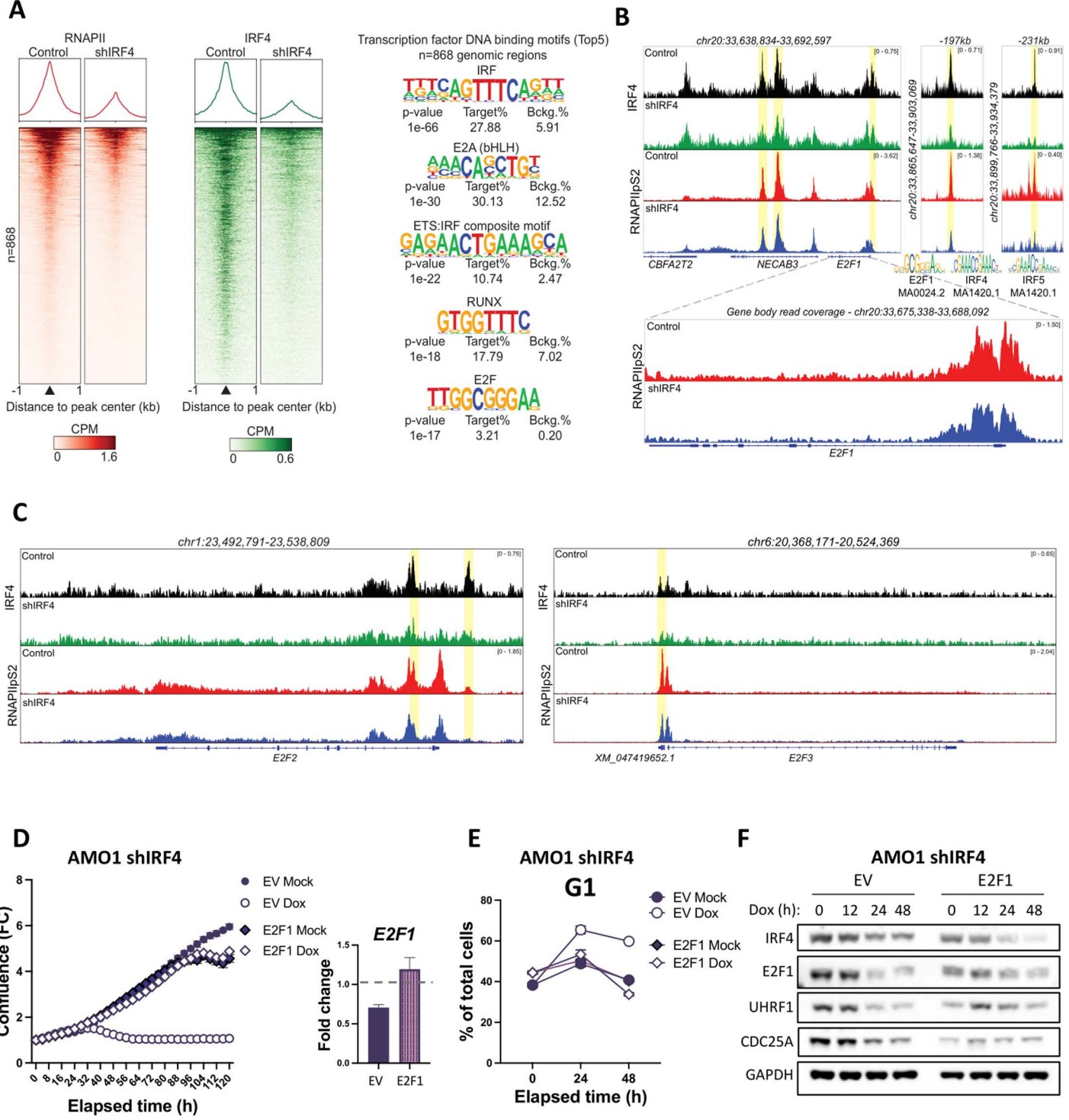

**Fig 5. IRF4 directly controls E2F1 gene transcription, which sufficiently regulates cell cycle downstream of IRF4.** (A) Differential binding analysis of IRF4 silencing by ChIP-seq. AMO1 shIRF4 Cl.1 cells were treated with or without Dox (0.2 μg/mL) for 24 h and then crosslinked and subjected to IRF4 and **RNAPIIpS2** ChIP-seq. Global differential binding analysis was performed for both RNAPIIpS2 and IRF4. Left: RNAPIIpS2 (red heatmap) and IRF4 (green heatmap) normalized read coverage (CPM - Counts per million) plotted for the 868 regions that were identified. *Right:* Top 5 Transcription Factor DNA binding motifs identified in the 868 regions by *de novo* motif discovery (Homer). Target% = percentage of site containing each motif. Bckg.% = percentage of background genomic regions that contain each motif. (B) Differential binding within E2F1 intra- and inter-genic regions. Normalized

counts from Fig 5A were visualized by genome browser. Each track depicts the intensity of IRF4 (black and green) or RNAPIIpS2 (red and blue) binding in IRF4 proficient or deficient cells respectively. The y-axis denotes signal intensity, while the x-axis indicates the genomic position. Yellow shadowing indicates the differential binding regions identified by the analysis in Fig 5A. JASPAR targeted transcription factor motif search against IRF or E2F1 motifs (75% relative profile score threshold) identified IRF4, E2F1, and IRF4 motifs in the E2F1-proximal regions. (C) Differential binding within E2F2 and E2F3 loci. Genome browser tracks depicting the intensity of IRF4 (black and green) or RNAPIIpS2 (red and blue) binding in IRF4 proficient or deficient cells. Yellow shadowing indicates the differential binding regions identified by the analysis in Fig 5A. (D) Rescue of *in vitro* spheroid growth of IRF4-deficient cells by ectopic E2F1 repletion. AMO1 shIRF4 cells were stably transfected with Dox-inducible empty vector (EV) or E2F1. AMO1 shIRF4 Ev or E2F1 Cl. 5 were cultured in the absence (closed symbols) or presence (open symbols) of Dox (0.2 µg/mL) and spheroid cell growth, depicted as FC confluence, was monitored by time-lapse microscopy in an IncuCyte instrument. (Right) Analysis of these cells after 24 h of culture in the same conditions by RT-qPCR for *E2F1* mRNA levels. Dashed line represents the level of endogenous *E2F1* expression. All values represented as mean ± SEM. (E) Effect of E2F1 repletion on G1 cell cycle arrest upon IRF4 silencing. AMO1 shIRF4 Cl.1 + EV or shIRF4 Cl.1 + E2F1 Cl. 5 cells were incubated in the absence or presence of Dox (0.2 µg/mL) for the indicated time, and stained with PI before flow cytometry analysis. Data presented as % of cells in G1 phase ± SEM. (F) Effect of E2F1 repletion on UHRF1 and CDC25A proteins in IRF4-deficient cells. AMO1 shIRF4 Cl.1 + EV or shIRF4 Cl.1 + E2F1 Cl. 5 cells were cultured in the absence or presence of Dox (0.2 µg/mL) for 24 h and analyzed by IB for IRF4, E2F1, UHRF1, CDC25A, and GAPDH as loading control. Data underlying this figure can be found in GEO archive (GSE288671), S2 Data, S1 Data, and S1 Raw images.

contrast, none of the 868 sites corresponds to the *RB1* locus (S5C Fig)—a gene that was not regulated by IRF4 (Fig 4C). We identified differential binding regions also within the *IRF4* locus itself (S5B Fig), providing additional evidence of IRF4 self-regulation. Corroborating our RNA-seq results (Fig 4), the ChIP-seq data indicated binding of IRF4 and RNAPIIpS2, both within and in proximity to the *E2F1* locus (Fig 5B). Given that E2F1 also exerts positive autoregulation [50–52], we explored whether the E2F1-proximal binding sites could control the E2F1 locus. To this end, we performed a targeted transcription-factor motif search using the JASPAR algorithm [53]. In these proximal sites, JASPAR identified not only IRF4 motifs but also an E2F1 consensus sequence (Fig 5B, middle). Thus, the *E2F1*-proximal binding site for IRF4 may represent an enhancer region that controls *E2F1* transcription. The ChIP-seq analysis also indicated IRF4 binding to *E2F2* and *E2F3* loci (Fig 5C), suggesting that IRF4 controls multiple E2F family members. Similarly, IRF4 regulated the loci of the known E2F transcriptional targets *Cdc25A* [48] and *UHRF1* [54–56] (S5D–E Fig), consistent with our previous findings (Fig 4C–4D and S3P Fig), and with cell-cycle regulation by IRF4 (Figs 3D–E and 4 and S3L Fig).

Further substantiating the importance of E2F1 among IRF4's transcriptional targets, ectopic E2F1 repletion in IRF4-depleted cells was sufficient to fully rescue growth and cell-cycle progression (Fig 5D–F). Interestingly, E2F1 repletion by itself rescued UHRF1 and CDC25A protein levels during IRF4 silencing (Fig 5F), suggesting that, although IRF4 can target these genes through direct binding, in this setting it controls them primarily via E2F1.

### IRF4 repletion in IRE1-deficient cells rescues E2F1 expression and proliferation

The regulation of IRF4 activity and abundance by IRE1 as well as the similarity between the cell-cycle phenotypes of IRE1 and IRF4 silencing supported the possibility of functional linkage between IRE1, IRF4, and cell-cycle control. To more specifically affirm IRF4's ability to circumvent nonenzymatic IRE1 dependency, we attempted to ectopically express IRF4 in AMO1 cells harboring inducible IRE1 silencing. Aiming for ectopic IRF4 levels that would not greatly exceed the endogenous amount [19,57], we used Dox-inducible expression of IRF4 (Fig 6A–6B and S6A Fig). Providing functional validation of this strategy, IRF4 re-expression successfully rescued shRNA silencing of endogenous IRF4 (S6A Fig). Interestingly, in the context of IRF4 knockdown, higher ectopic IRF4 levels did not enable a better rescue (S6A Fig, compare clones 4 and 5). This result suggests that IRF4 does not act merely as a global driver of MM proliferation. Consistent with this interpretation, ectopic IRF4 expression in cells expressing non-targeting control shRNA did not provide a further proliferative boost (S6B Fig). Crucially, IRF4 repletion was sufficient to rescue proliferation of AMO1 cells in the context of IRE1 silencing (Fig 6A–6B). Notwithstanding, although IRF4 re-expression effectively restored proliferation for up to 72 h, a modest growth decline of up to 15% occurred beyond this time point (Fig 6B). Of relevance, IRF4 repletion only partially prevented apoptosis induction during IRE1 silencing (Fig 6C and S6C Fig). This residual, non-IRF4 dependent apoptosis induction may be attributed to the upregulation of Death Receptor 5 (DR5) (S6D Fig)—a previously established RIDD target [58–60]—in

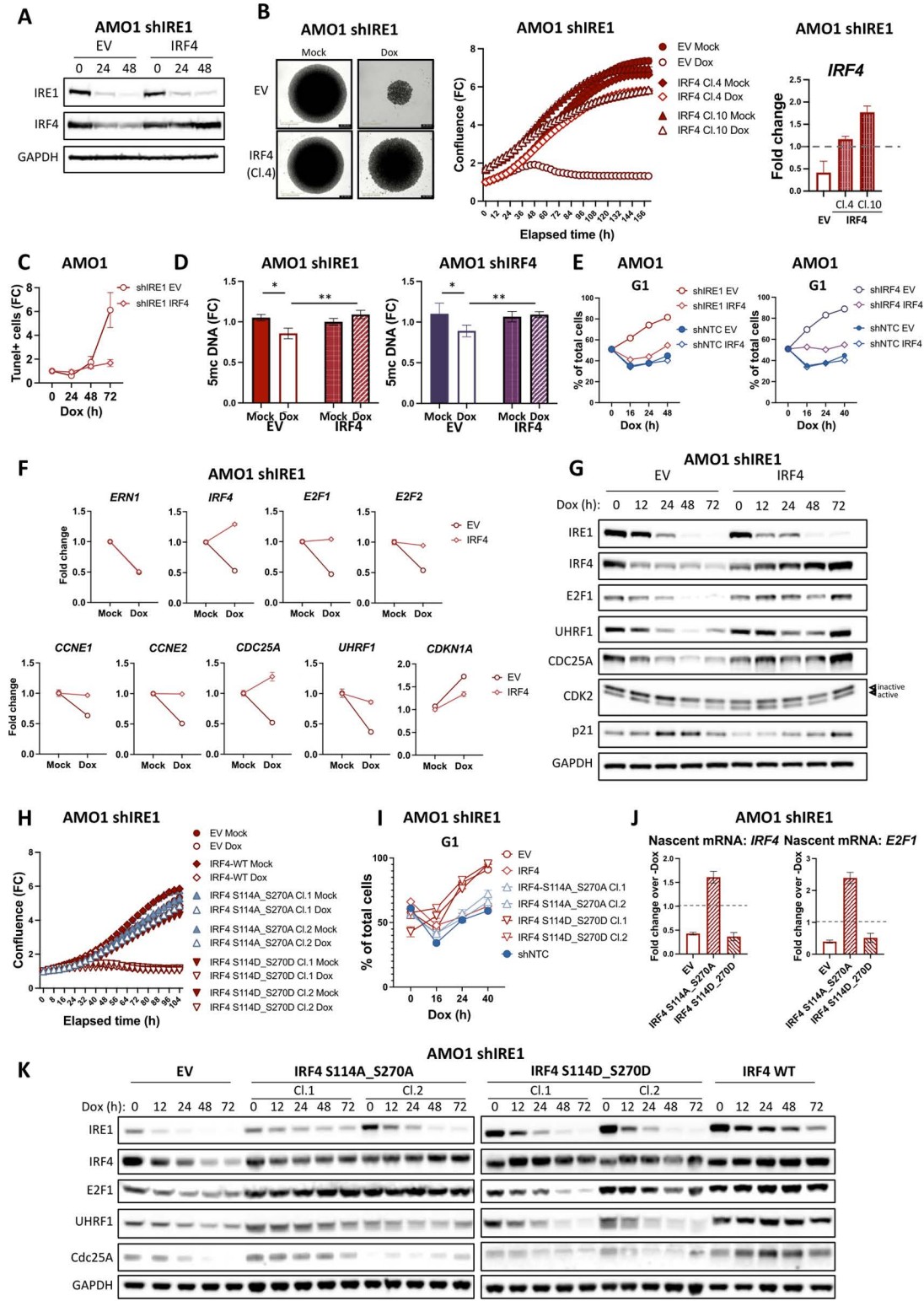

**Fig 6. IRF4 repletion in IRE1-deficient cells rescues E2F1 expression and proliferation.** (A) Dox-inducible ectopic IRF4 expression in AMO1 shIRE1 cells. AMO1 shIRE1 Cl.1 cells were stably transfected with a Dox-inducible IRF4 or with empty vector (EV) as a control. AMO1 shIRE1 EV or shIRE1 IRF4 Cl.4 cells were treated with Dox (0.2 μg/mL) for 0, 24 or 48 h and analyzed by IB for IRF4 protein levels as well as IRE1 protein depletion.

(B) Rescue of *in vitro* spheroid growth of IRE1-deficient cells by IRF4 repletion. AMO1 shIRE1 EV or 2 individual shIRE1 IRF4 clones were cultured in the absence (closed symbols) or presence (open symbols) of Dox (0.2 μg/mL) and spheroid cell growth was monitored by time-lapse microscopy in an IncuCyte instrument. *Left*: representative images of indicated clones after 7 days of culture as single spheroids. *Middle:* Spheroid growth, depicted as FC confluence. *Right:* Analysis of these cells after 24 h of culture in the same conditions by RT-qPCR for *IRF4* mRNA levels. Dashed line represents the level of endogenous *IRF4* expression. All values represent mean ± SEM. (C) Effect of IRF4 repletion on apoptosis induction by IRE1 silencing. AMO1 shIRE1 Cl.1 EV or shIRE1 Cl.1 IRF4 Cl. 4 cells were incubated in the absence or presence of Dox (0.2 μg/mL) for the indicated time and subjected to TUNEL assay before analyzed by flow cytometry. Representative biological replicate out of 3. Values represented as mean ± SEM. (D) Effect of IRF4 repletion on DNA demethylation induced by IRE1 silencing. AMO1 shIRE1 Cl.1 EV or shIRE1 Cl.1 IRF4 Cl. 4 and shIRF4 Cl.1 EV or shIRF4 Cl.1 IRF4 Cl. 5 cells were incubated in the absence or presence of Dox (0.1 μg/mL) for the indicated time and DNA was extracted. DNA methylations were then analyzed by 5mc-DNA ELISA. Due to the minimal amount of DNA methylations in AMO1 cell line (approximately 0.1%), the results are presented as FC to the parental cell line, based on three independent biological replicate means ±SEM. (E) Effect of IRF4 repletion on IRE1 silencing-induced G1 cell cycle arrest. AMO1 shIRE1 Cl.1 EV or shIRE1 Cl.1 IRF4 Cl. 4 and shIRF4 Cl.1 EV or shIRF4 Cl.1 IRF4 Cl. 5, as well as shNTC +EV and shNTC IRF4 Cl. 3 cells were incubated in the absence or presence of Dox (0.1 μg/mL) for the indicated time, EtOH-fixed and stained with PI before they were analyzed by flow cytometry. Data presented as % of cells in G1 phase from a representative experiment. (F) Effect of IRF4 repletion on E2Fs and E2F targets during IRE1 depletion. AMO1 shIRE1 Cl.1 EV or shIRE1 Cl.1 IRF4 Cl. 4 cells were cultured in the absence or presence of Dox (0.1 μg/mL) for 24 h and analyzed by RT-qPCR for *E2F* mRNAs and other targets. Values represented as mean ± SEM. (G) Effect of IRF4 repletion on E2F1 protein in IRE1 deficient cells. AMO1 shIRE1 Cl.1 EV or shIRE1 Cl.1 IRF4 Cl. 4 cells were cultured in the absence or presence of Dox (0.1 μg/mL) for a time course up to 72 h and analyzed by IB for E2F1, UHRF1, CDC25A, CKD2, and p21. (H) Rescue of *in vitro* growth of IRE1-deficient cells by WT or phospho-mutant IRF4 repletion. AMO1 shIRE1 EV or 2 individual shIRE1 IRF4 S114A_S270A or S114D_S270D clones were cultured in the absence (closed symbols) or presence (open symbols) of Dox (0.2 μg/mL) and spheroid cell growth, depicted as FC confluence, was monitored by time-lapse microscopy in an IncuCyte instrument. Values represented as mean ± SEM. (I) Effect of phospho-mutant vs. WT IRF4 repletion on G1 cell-cycle arrest induction by IRE1 silencing. AMO1 shIRE1 Cl.1 EV or IRF4 Cl. 4 or S114A_S270A Cl. 1; 2 or S114D_S270D Cl. 1; 2 cells were incubated in the absence or presence of Dox (0.2 μg/mL) for the indicated time points, EtOH-fixed and stained with PI before they were analyzed by flow cytometry. Data presented as % of cells in G1 phase from a representative experiment. (J) Analysis of *IRF4* and *E2F1* mRNA transcription by nuclear run-on. AMO1 shIRE1 Cl.1 cells expressing Dox-inducible EV or IRF4 phosphomutants were treated with 0.2 μg/mL Dox for the indicated time. Nascent RNA was labeled with a uridine analogue, captured and analyzed for *IRF4* and *E2F1* by RT-qPCR. Data represented as mean ± SEM. Dashed lines represent the levels of expression in the mock (absence of Dox) samples. (K) Effect of phosphomutant IRF4 repletion on E2F1 protein in IRE1 deficient cells. AMO1 shIRE1 Cl.1 + EV or shIRE1 Cl.1 + S114A_S270A Cl. 1; 2, or shIRE1 Cl.1 S114D_S270D Cl. 1; 2, or shIRE1 Cl.1 IRF4 Cl. 4 cells were cultured in the absence or presence of Dox (0.2 μg/mL) for a timecourse up to 72 h and analyzed by IB for E2F1, UHRF1, and CDC25A. Data underlying this figure can be found in S1 Data and S1 Raw images.

the absence of IRE1 and RIDD. Indeed, as expected, IRF4 repletion did not prevent DR5 upregulation upon IRE1 knockdown (S6D Fig), in keeping with IRF4's inability to recover IRE1's enzymatic activity. It is therefore likely that residual caspase activation under IRF4 repletion limits growth restoration beyond 72 h. Of note, IRF4 repletion was also sufficient to rescue the previously documented decrease in DNA methylation due to IRE1 silencing [15] (Fig 6D). Importantly, IRF4 re-expression enabled a substantial recovery of G1/S transition in IRE1-depleted cells (Fig 6E). Furthermore, it rescued the expression of *E2F1*, *UHRF1*, and *CDC25A* and restored CDK2 activity (Fig 6F–G and S6E–F Fig).

The phenotypic similarity between the silencing of IRE1 and IRF4, together with the regulation of IRF4 by IRE1, and the ability of IRF4 to recover cell-cycle progression upon IRE1 silencing, supported the possibility that IRF4 operates downstream to IRE1, rather than in parallel. The IRE1-controlled phosphorylation sites on IRF4 would more likely be important for rescuing proliferation if IRE1 and IRF4 operated in linear, rather than parallel fashion. Corroborating our earlier results (Fig 2H), the phospho-deficient mutant of IRF4 showed stronger chromatin binding than did WT IRF4, whereas the phospho-mimetic mutant displayed little or no binding (S6H Fig). Strikingly, the phospho-deficient mutant rescued proliferation and cell-cycle progression of IRE1-depleted cells similarly to WT IRF4, whereas the phospho-mimetic mutant failed to recover these cellular outcomes (Fig 6H–6I). Furthermore, the two phosphorylation-site mutants also showed the same relative behavior as regards to IRF4's transcriptional activity, evident by the regulation of the endogenous *IRF4* as well as *E2F1* and *PRDM1* nascent mRNAs (Fig 6J and S6I Fig). Moreover, the phospho-deficient or WT IRF4 variants, but not the phospho-mimetic mutant, rescued the expression of IRF4, E2F1, UHRF1, and Cdc25A at both the RNA and protein levels (Fig 6F and 6G, 6K and S6J Fig). Of note, the phospho-deficient and WT variants, but not the phospho-mimetic mutant, reversed the decrease in chromatin binding by Myc upon IRE1 silencing (S2C and S6H Figs), indicating that in these cells IRE1 also regulates Myc through IRF4's phosphorylation state. Together, these results demonstrate that IRF4 repletion in

the context of IRE1 silencing reverses all the primary antiproliferative features of IRE1 depletion. Furthermore, the rescue depends on IRE1-controlled IRF4 phosphorylation sites, indicating that in these IRE1-dependent MM cells, IRF4 operates downstream of IRE1 to promote cell-cycle progression.

## Discussion

Our study highlights the pan-MM essential transcription factor, IRF4, as a crucial mechanistic conduit of the recently discovered nonenzymatic dependency on IRE1 in MM cells (Fig 7). We show that in the context of nonenzymatic IRE1

**IRE1-proficient cell**

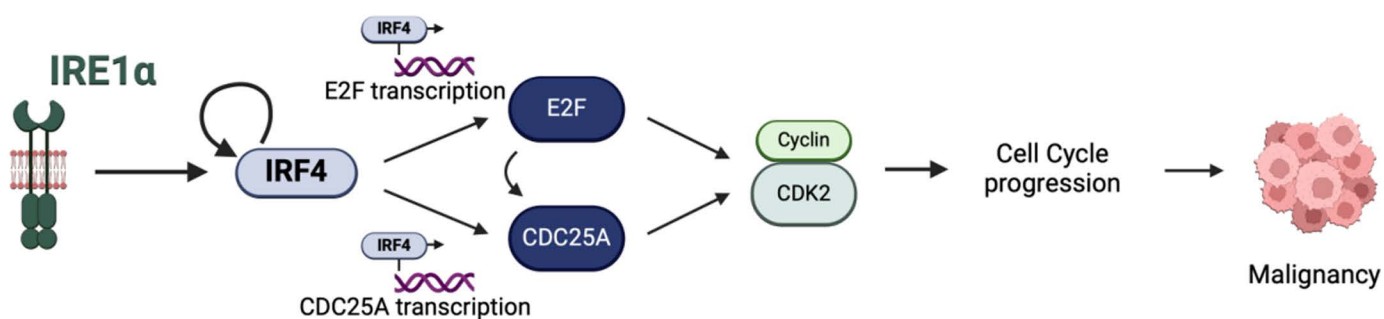

**IRE1-deficient cell**

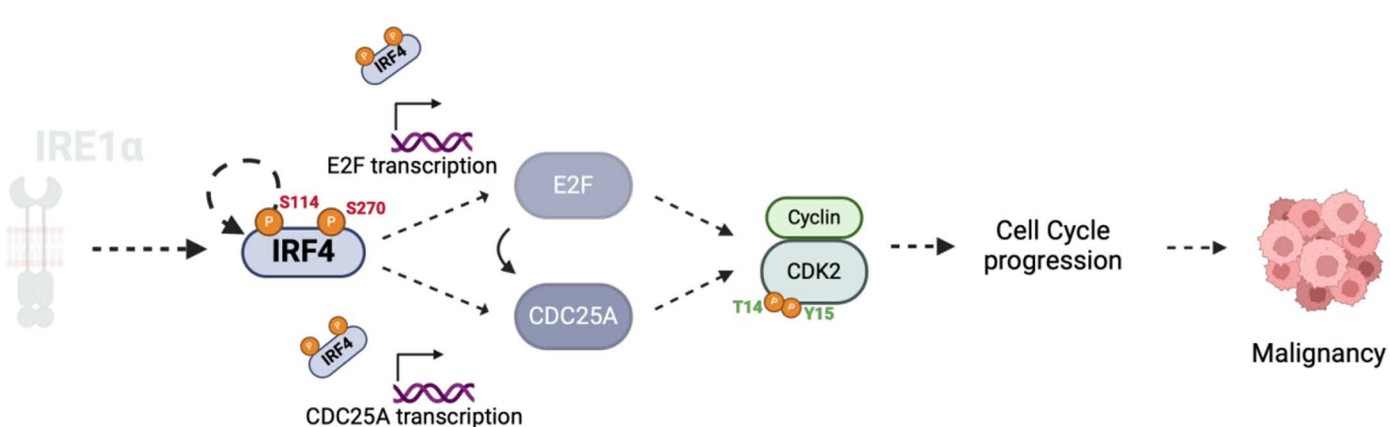

**Fig 7. Hypothetical model for nonenzymatic IRE1 cell cycle regulation via IRF4.** Schematic representation of IRE1 supporting IRF4 activity/ expression, promoting E2F1 and CDC25A transcription, in turn driving CDK2 activity to enable unconstrained cell-cycle progression. Silencing of IRE1 increases IRF4 phosphorylation on Ser114 and Ser270, which attenuates IRF4's self-sustaining transcriptional activity, depleting E2F1 and CDC25A and leading to cell-cycle arrest. Phosphoresidues annotated in red: inhibitory. Phosphoresidues annotated in green: activating. Created in BioRender. Oikonomidi, I. (2025) https://BioRender.com/b50x169.

dependency, IRE1 sustains *IRF4* transcription independently of XBP1s. In contrast, in the context of enzymatic IRE1 dependency, IRE1 can support IRF4 levels via XBP1s. Furthermore, in IRE1-independent cells, IRE1 does not appear to play a pivotal role in IRF4 regulation. Thus, the underlying type of IRE1 requirement in a given MM cell line seems to govern IRF4 regulation by IRE1.

Our results further indicate that in cell lines that require nonenzymatic IRE1 function, IRF4 acts primarily as a self-sustaining driver rather than being controlled by other transcription factors such as Myc (S7 Fig). Moreover, we demonstrated that IRE1 nonenzymatically supports IRF4 expression by modulating specific phosphorylation sites on the IRF4 protein. IRE1 silencing significantly increased IRF4 phosphorylation on Ser114 and Ser270—a previously unknown modification that markedly inhibited chromatin binding by IRF4. We further showed that this phosphorylation inhibits IRF4's transcriptional activity both toward itself and toward known target genes.

Our further mechanistic investigation revealed that IRF4 engages the *E2F1* and *CDC25A* genes and promotes CDK2 activation to drive cell-cycle progression (Fig 7). These observations lay the groundwork for future investigations aiming to identify the specific scaffolding role of IRE1 that regulates IRF4 phosphorylation and activity, thereby controlling MM cell-cycle progression. Phosphatases or kinases that have been previously linked to IRE1 [61] could provide interesting candidates.

Our functional and transcriptomic studies demonstrated an extensive phenotypic similarity between silencing of IRE1 and IRF4. Crucially, IRF4 re-expression recovered cell-cycle progression and proliferation, as well as E2F1 expression, UHRF1 and Cdc25A abundance, and CDK2 activity in the context of IRE1 knockdown. Taken together with the regulation of IRF4 activity and abundance by IRE1, and the dependency of the rescue by IRF4 on its IRE1-controlled phosphorylation sites, these results strongly support the conclusion that IRF4 acts downstream to, rather than independent of, IRE1 to control IRE1-dependent MM growth. Of note, IRF4 re-expression did not prevent apoptosis induction upon IRE1 silencing—likely due to the upregulation of DR5 in the absence of RIDD. Nevertheless, apoptosis was dispensable for cell-cycle regulation by IRE1 [15].

Our findings further suggest that E2F1 plays a key role in mediating cell-cycle control downstream of IRE1 and IRF4: (i) the prominence of E2F targets among DEGs common to IRE1 and IRF4; (ii) the ChIP-seq results identifying *E2F1* as a direct target of IRF4; and (iii) the ability of E2F1 to rescue growth inhibition upon IRF4 silencing. To our knowledge, this is the first example of successful functional substitution of IRF4 by one of its transcriptional targets. E2F1 repletion also helped sustain Cdc25A levels, suggesting that IRF4 regulates this phosphatase both through direct enhancer engagement and indirectly via E2F1. The extended loss of E2F1, as well as of CDC25A—a central CDK2-activating phosphatase [62]—may explain the inactivation of CDK2 upon IRE1 or IRF4 silencing. In turn, CDK2 inactivation facilitates G1 arrest or faster G2/M resolution [43]. Regardless, CDK2 activity proved to be a reliable functional biomarker for IRE1 or IRF4-mediated cell-cycle regulation.

The identification of IRF4 as a critical conduit of nonenzymatic IRE1-dependent cell-cycle control in MM has important potential translational implications for the development of novel diagnostic and therapeutic strategies for both targets. It would be valuable to further explore the interplay between IRE1 and IRF4 in the context of immune cells and anti-cancer immunity. Recent evidence supports IRE1 disruption as a potential strategy to enhance anti-tumor immunity [4,63], while cytotoxic immune responses are pivotal in refractory MM, in which IRF4 becomes dispensable [64]. To our knowledge, this study represents the first *mechanistic* investigation of cell-cycle regulation by IRF4, and thereby advances the current functional understanding not only of IRE1, but also IRF4 in MM.

## Materials and methods

### Reagents

Antibodies: XBP1s [10,59], IRE1α LD [65], pIRE1 [10,59] were from Genentech. GAPDH-HRP (#2118), IRF4 (#4964), MITF (#12590), Aiolos (#12720), Cofilin-HRP (#8503), Histone H3 #9715), Lamin B2 (#12255), Ubiquitin (#3936), CDK2 for immunoprecipitation (#18048), Rb (#9309), p21 (#2947), actin-HRP (#5125), Lamin A/C (#2032), PARP1 (#9532), Caspase 3 (#9662), Caspase 8 (#9746),

p53 (#18032), pT160 CDK2 (#2561), CDK1 (#9116), CDK6 (#13331), Cyclin E2 (#4132), Cyclin B1 (#4138), p27 (#2552), pS608 Rb (#2181), pS795 Rb (#9301), PERK (#5683), eIF2α #9722), and CHOP (#2895) were from Cell Signaling Technology (CST). CD59 (ab133707) c-Myc (ab32072), CDK2 (ab32147), pT14 CDK2 (ab68265), pT124 Cdc25A (ab182666), and Cdc25A (ab2357), Histone H2AX (ab20669), and RNAPII-pS2 (ab5095) were from Abcam. γH2AX (#05−636) was from Millipore, pT24 CDK6 (#PA537518) from Thermo Fisher Scientific, while E2F1 (#66515-1-Ig), ATF6 (#66563–1) and UHRF1 (#21402-1-AP) were from Proteintech.

IRE1 Inhibitors: Kinase inhibitor (KI) (1−3 µM, Ref. [66]), RNase inhibitor (RI) 4µ8C (10 µM, aka IRE1 inhibitor III, Ref. [67]) and RNase inhibitor (RI2; 1 µM; aka MKC8866, Ref. [68]) were from Genentech. CDK1 inhibitor RO-33306 (9 µM, Ref. [69]) was from MedChem Express. Pan-Caspase inhibitor Q-VD-Oph (50 µM; Q-VD) was from SelleckChem. Doxy-cycline (0.1–0.2 µg/mL; Dox) was from Clontech. Etoposide (25 µM, #1226) was from R&D systems. Actinomycin D (1 µg/mL, #11805017) was from Gibco.

## Corrected gene-effect score calculation

CrisprGeneEffect files were downloaded from DepMap, Broad (2023). DepMap 23Q4 Public Dataset. https://doi.org/10.25452/figshare.plus.24667905.v2. An average gene effect score was calculated for all tested cell lines (1,101) which was, then, subtracted by the Chronos score of each gene in the multiple myeloma cell lines of interest (AMO1, KMS27, L363, JJN3). This correction allowed for removing pan-essential genes and focusing on MM-specific dependencies. Further, by filtering for genes essential (≤ −0.25) in all nonenzymatically IRE1-dependent cell lines, we generated a list of 131 genes (S1 Table) that are nonenzymatic IRE1 co-dependencies.

## Cell culture

All cell lines were obtained or generated from an internal repository maintained at Genentech. Multiple Myeloma cell lines (AMO1 (RRID:CVCL_1806), KMS27 (JCRB Cat# JCRB1188, RRID:CVCL_2993), L363 (RRID:CVCL_L363), KMS11 (RRID:CVCL_2989), NCI-H929 (RRID:CVCL_1600), and SR786 (SR; RRID:CVCL_1711)) were maintained under standard conditions in RPMI 1640 containing 2 mM glutaMAX, 100 U/mL penicillin, 100 µg/mL streptomycin (all from Gibco), supplemented with 10% fetal bovine serum (Sigma).

## Cell line authentication/quality control

Short tandem repeat (STR) Profiling: STR profiles are determined for each line using the Promega PowerPlex 16 System. This is performed once and compared to external STR profiles of cell lines(when available) to determine cell line ancestry. Loci analyzed: Detection of 16 loci (15 STRloci and Amelogenin for gender identification), including D3S1358, TH01, D21S11, D18S51, Penta E, D5S818, D13S317, D7S820, D16S539, CSF1PO, Penta D, AMEL, vWA, D8S1179, and TPOX. Cell lines were also tested to ensure they were mycoplasma-free within 3 months of use.

## Generation of engineered cell lines

Construction of AMO1, KMS27 or L363 shIRE1, shXBP1 or shNTC cells was described previously [8]. Cells were transfected simultaneously with three different shRNAs for either IRF4 or with a non-targeted control shRNA, encoded in a puromycin resistance piggyBac vector, controlled by a doxycycline-inducible promoter. Transfected cells underwent puromycin selection and single-cell cloning when required. Same for SR786 shIRE1 and H929 shIRE1/ shNTC and AMO1 shIRF4, AMO1 shMYC, KMS27 shIRF4, and KMS27 shNTC, Clones with the most complete knockdown levels were selected by immunoblot and qPCR.

*IRF4* shRNAs:
5′-TTGAAGAAGCCTCACACGTAA-3′, 5′- CAGGATGGAGCTGACTACGGA-3′,
5′-AGGATGGAGCTGACTACGGAA-3′.
*MYC* shRNAs: 5′-TAGTCGAGGTCATAGTTCCTG-3′, 5′-TCGGTCACCATCTCCAGCTGG-3′,

5′-TACGGCTGCACCGAGTCGTAG-3′.

Full-length human IRF4 (wildtype or mutated) or E2F1 were cloned into a Doxycycline-inducible promoter in a piggy-Bac plasmid encoding for Blasticidin resistance. AMO1 shIRE1 Cl.1 or AMO1 shIRF4 Cl.1 cells were transfected with IRF4 inducible plasmid using Mirus TransIT-X2 delivery system. Blasticidin-selected cells were single-cell cloned and clones were screened for IRF4 or E2F1 induction levels by qPCR and immunoblotting. The mutated IRF4 constructs were codon optimized so that they are not recognized by endogenous IRF4 Taqman primers hence a custom primer had to be used.

## siRNA transfections

siRNAs against *MYC* were purchased by Dharmacon Reagents. $5 \times 10^6$ KMS27 cells were electroporated with 2 µM siMYC, using the Nucleofector Kit V (Lonza) according to the manufacturer's protocol and program X-001. Cells were incubated for 24 h, 48 h, and 72 h and collected for RNA and protein extraction as described above.

## Apoptosis assays

**CaspaseGlo.** Equal number of cells was seeded in opaque 96well plates and equal volume of Caspase 3/7 Glo (Promega) was added according to the manufacturer's instructions. Luminescence was measured using a PerkinElmer reader.

**AnnexinV/PI staining.** Dead Cell Apoptosis kit (V13242, Invitrogen) was used according to the manufacturer's instructions. In brief, past the indicated treatments, $1 \times 10^6$ cells were stained with FITC Annexin V and PI according to the manufacturer's instructions and visualized on a FACSymphony cell analyzer (BD). Percentages of cells in early (Annexin V + PI -) or late (Annexin V + PI +) apoptosis were determined with FlowJo10 software.

**Tunel assay.** DNA fractionation in AMO1 shIRE1 EV versus shIRE1 IRF4 cl.4 cells were measured using the In Situ Cell Death Detection Kit, Fluorescein (Roche, 11684795910) according to protocol. In brief, 2 million cells per treatment were stained with LIVE/DEAD fixable dye (Life Technologies, 30 min at RT) and then fixed in 2% PFA solution. After permeabilization, cells were treated with TUNEL reaction mixture. A positive control was treated with DNase I solution prior to TUNEL reaction. Immediately after labeling, the samples were analyzed on a FACSymphony cell analyzer (BD). After excluding dead cells, apoptotic cells were determined using FlowJo 10.10.0 software.

## RT-qPCR

As described before [10], RNA was extracted with RNeasy Plus kit (Qiagen). Equal amounts of RNA were reverse transcribed and amplified by RNA-to-Ct kit (Applied Biosystems) on the ViiA 7 Real-Time PCR System. The δδCt values were calculated by relating each individual Ct value to its internal GAPDH control, and then normalizing to the vehicle-treatment control, using the ViiA 7 software. Taqman primers (Life Technologies): *GAPDH*: Hs02758991_g1; *XBP1u*: Hs02856596_m1; *XBP1s*: Hs03929085_g1; *DGAT2*: Hs01045913_m1; *CD59*: Hs00174141_m1; *endogenous or wildtype IRF4*: Hs00180031_m1; S114A_S270A and S114D_S270D *IRF4*: APRWP97; *ERN1*: Hs00980095_m1; *MITF*: Hs01117294_m1; *MYC*: Hs01031255_m1; *IKZF1*: Hs00958474_m1; *IKZF3*: Hs05037772_s1; *Cdc25A*: Hs00947994_m1; *E2F1*: Hs00153451_m1; *E2F2*: Hs00918090_m1; *CDK1*: Hs00938777_m1; *CDK2*: Hs01548894_m1; *CDK4*: Hs00364847_m1; *CDK6*: Hs01026371_m1; *CDKN1A*: Hs00355782_m1; *CCND2*: Hs00153380_m1; *CCNA2*: Hs00996788_m1; *RB1*: Hs01078066_m1; *CCNE1*: Hs01026536_ m1; *CCNE2*: Hs00180319_m1; *UHRF1*: Hs01086727_m1; *PRDM1*: Hs00153357_m1; *ATM*: Hs00175892_m1, *CCNB3*: Hs00364460_m1; *CCNB1*: Hs99999188_m1.

## Immunoblot

Cells were washed in ice-cold PBS and lysed in RIPA lysis buffer (20–188, Millipore) supplemented with cOmplete protease inhibitor cocktail (Roche) and PhosSTOP (Roche) tablets on ice for 30 min. Clarified lysates

(16,100 g for 10 min at 4 °C) were quantified by BCA protein assay (Thermo Fisher Scientific) and normalized lysates were denatured with 3x NuPAGE LDS (or SDS where indicated) buffer supplemented with 0.4M Dithiothreitol (DTT, Invitrogen) at 95 °C for 5 min. NuPAGE pre-cast gels with MOPS running buffer (or Tris-Glycine Wedgwell pre-cast gels with Tris-Glycin running buffer, where indicated) (Invitrogen) were used for SDS-PAGE electrophoresis and protein was electro-transferred to nitrocellulose membranes using the iBLOT dry blotting systems (Invitrogen). Membranes were then blocked in 5% non-fat milk solution for 45 min at RT and probed with the corresponding primary antibodies overnight at 4 °C. Following appropriate washing, blots were incubated with secondary (HRP-conjugated) antibodies (Jackson Immunoresearch Laboratories; 1:10,000 dilution) for 1 h at RT. HRP Chemiluminescence was measured in an Azure imaging system (Azure Biosystems) using SuperSignal West Dura Extended Duration Substrate or SuperSignal West Pico PLUS Chemiluminescent Substrate (Thermo Fisher Scientific).

### Subcellular fractionation

Subcellular fractionations were done using the Subcellular Protein Fractionation Kit for Cultured Cells (Thermo Scientific). Cells were pelleted and washed in PBS. The pellets were sequentially lysed with different lysis buffers containing 1x Halt Protease Inhibitor (Thermo). Each post-lysis supernatant was a different cellular fraction (i.e., Cytoplasmic (C), Membrane (M), Soluble nuclear (or Nuclear envelope; SN), Chromatin-bound nuclear (CN)). Lysates were then denatured with NuPAGE LDS supplemented with Dithiothreitol (Invitrogen) at 95°C for 5 min and immunoblots followed. Quantification of chromatin binding was performed by densitometry, followed by normalization of each CN band to SN band (mentioned as binding ratio). Then, comparison to a Mock or WT-expression condition resulted in the Fold change values presented in the graphs.

### Densitometry

Semi-quantitative densitometric analysis on chemiluminescence images from immunoblot exposures was performed with Fiji software, measuring at least three independent experiments.

### Confluence

3D proliferation assays were described previously. In brief, 5,000 cells per well were plated in ULA 96-well plates (Corning) and single spheroids were formed by centrifugation (600 g) for 5 min. Treatments were used at the indicated concentrations at the time of plating. Spheroids were imaged in an incubator of 37°C, 5% $CO_2$, and 95% humidity using an IncuCyte instrument (EssenBioscience). Frames were captured at 4-h intervals using a 4× objective and confluence (%) or spheroid µm²/image were calculated on the instrument. For consistent representation, results are presented as fold change (FC) to the initial frame.

### Nuclear run-on

For the isolation of nascent RNA chains from AMO1 shIRE1 cl.1 cells Click-iT Nascent RNA Capture Kit (Life Technologies) was used. Cells were first incubated with 0.2 µg/mL Doxycycline for the indicated durations and 1 h before the end of the Dox incubation, the cells were pulsed with 0.5 mM EU for 1 h and total RNA was extracted using RNeasy Plus Mini kit (Qiagen). 1 µg RNA input was biotinylated with a Click-it reaction according to the manufacturer's instructions and precipitated with Glycogen provided in the kit and GlycoBlue Coprecipitant (Thermo Fisher Scientific) to help visualization for 2.5 hours at −80 °C before treated with DNAse and captured on Dyna beads. Biotinylated (nascent) RNA was reverse transcribed using SuperScrip IV VILO Master Mix (Thermo Fisher Scientific) immediately. Following, quantification of cDNA was achieved by qPCR, using the TaqMan Fast Advanced Master Mix for qPCR (Thermo-Fisher

Scientific) on a ViiA 7 Real-Time PCR System (Thermo-Fisher Scientific) based on Ct values relative to GAPDH internal control.

## Co-immunoprecipitations

Cells were lysed for 30 min on ice in NP-40 lysis buffer (25 mM HEPES, pH 7.4, 150 mM NaCl, 1 mM EDTA, 1% NP-40, 10% glycerol) containing cOmplete protease inhibitor cocktail and PhosSTOP tablets (Roche). Post-nuclear supernatants were pre-cleared with unconjugated magnetic beads (Thermo-Fisher Scientific) or agarose at 4°C for 60 min with rotation, followed by capture on magnetic beads plus CDK2 antibody (1:100) for 90 min. Beads were washed 3–5 times, for 10 min, at 4°C in the same NP-40 lysis buffer supplemented with NaCl to 300 mM. Samples were eluted with 1.5 × SDS-PAGE sample buffer and incubated at 95°C for 5 min before loading.

## Cell proliferation (CFSE)

Cells were incubated with 5 µM Cell Trace Violet (Thermo-Fisher Scientific). After a 20-minute incubation in a 37°C water bath, Serum-Free RPMI 1640 was added to the cells and incubated for 5 min to absorb unbound dye. Cells were pelleted and plated -/+ Dox (or 25 µM Etoposide as a negative proliferation control) for 72 h. Cells were, then, harvested and stained with LIVE/DEAD Fixable Dead Cell Stain Kit (Life Technologies), and analyzed immediately using a FACSymphony analyzer (BD).

## DNA replication

DNA replication was assessed by BrdU incorporation using the eBioscience BrdU Staining Kit for Flow Cytometry (Invitrogen). Cells were treated with Doxycycline or Etoposide (negative proliferation control) in triplicate. BrdU in a final concentration of 10 µM was added and the cells were returned back inside the incubator for 45 min. Cells were collected and stained with the Zombie Fixable Viability Kit (Biolegend) before fixating and permeabilizing with BrdU staining working solution, followed by DNase I treatment. Finally, anti-BrdU antibody was added or IgG isotype control and the samples were analyzed using a FACSymphony analyzer (BD).

## G2/M cell cycle synchronization and sorting

Cells were treated with 9 µM CDK1i (RO-3306) for 21 h in order to block cell cycle (G2/M phase). Cells were then stained with live-cell DNA stain: Vybrant DyeCycle Violet Stain (Thermo Fisher Scientific), and sorted according to their DNA content on a BD Fusion/S6 sorter. G2/M cells were returned back to culture. Before the end of the incubation 50 µM EdU (Click-iT EdU Flow Cytometry, Invitrogen) were added in the cultures for 40 min. Cells were then collected and stained with LIVE/DEAD Fixable Dead Cell Stain Kit (Invitrogen) for 30 min on ice before proceeding with fixation/permeabilization and Click-it EdU reaction. Finally, DNA content was stained in saponin perm/fix solution using Hoechst ready-Flow (Thermo Scientific). The samples were then analyzed using a FACSymphony analyzer (BD) and, using FLowJo 10.10.0, live cells were allocated into cell cycle phases.

## DNA methylation (5-mc ELISA)

Total DNA was extracted from $1 \times 10^6$ cells using the AllPrep DNA/RNA Mini Kit (Qiagen). The extracted RNA was used to verify by RT-qPCR the effectiveness of the annotated treatments. DNA was quantified using Nanodrop spectrophotometer (Thermo Fisher Scientific) and 100 ng were analyzed by MethylFlash Global DNA methylation (5-mC) ELISA Easy kit (Epigentek), following the manufacturer's protocol. Absorbance was read on a SpectraMax M2 Microplate Reader (Molecular Devices) at 450 nm and total 5mc-DNA was calculated based on the absorbance of the 5mc-DNA standard curve. Due to the minimal amount of DNA methylations in AMO1 cell line (approximately 0.1%), the results are represented as a fold change percentage.

## Phospho-proteomics

**Sample preparation.** AMO1 shIRE1 Cl.1 cells were harvested and lysed in HEPES buffer (20 mM, pH 8.0) containing 9 M urea and phosphatase inhibitors (1 mM sodium orthovanadate, 2.5 mM sodium pyrophosphate, and 1.0 mM ß-glycerophosphate). Lysates from 4 conditions were generated for phosphoproteomic analysis. The treatment conditions were DMSO, 0.2 µg/mL Doxycycline, IRE1 KI, and IRE1 RI2 for 24 h. Each condition was performed in 4 biological replicates, for a total of 16 samples. Lysates were sonicated followed by centrifugation at 20,000 g for 20 min at 15°C. Protein concentration was measured using Bradford assay (BioRad, Hercules, CA). Proteins were reduced with 5 mM dithiothreitol (DTT) at 37°C for 1 h followed by alkylation with 15 mM iodoacetamide (IAA) at room temperature (RT) for 20 min in the dark. Samples were diluted to a final concentration of 2 M urea prior to digestion with Lys-C (Wako, Japan) at an enzyme:substrate ratio (E:S) of 1:50 at 37°C for 2 h followed by trypsin (Promega, Madison, WI) digestion at an E:S ratio of 1:50 at 37°C overnight. The digest mixtures were acidified with 20% trifluoroacetic acid (TFA) prior to solid-phase extraction using C18 cartridge (100 mg absorbent) from Waters (Milford, MA). Peptides were eluted with 3 × 0.5 mL of 60% acetonitrile (ACN)/0.1% TFA followed by peptide concentration measurement using a quantitative colorimetric peptide assay kit (Thermo, San Jose, CA). Equal amounts from each condition (2 mg) were subjected to lyophilization overnight.

**Chemical labeling with TMTPro reagent.** The dried peptide mixtures were reconstituted in HEPES (100 mM, pH 8.5) to a concentration of 1 mg/mL. TMTPro reagent, channels 126–134N, was added to each of the 16 samples at a 1:1 TMT:peptide ratio and a final concentration of ~17% ACN. Labeling was performed at RT for 1.5 h. A small portion (2 µL) from each condition was mixed, desalted, and analyzed to determine labeling efficiency. The reaction was quenched with 100 µL of 5% hydroxylamine once labeling efficiency was determined to be at least 95%. Samples were mixed followed by acidification using 20% TFA and lyophilized overnight. The 16-plex TMT labeled peptide mixture was desalted using C18 cartridge (1 g absorbent) from Waters (Milford, MA). Peptides were eluted with 3 × 4 mL of 60% ACN/0.1% TFA. A small portion (500 µg) was lyophilized for proteome analysis so that during data analysis we can normalize phosphorylation level against protein level. The rest of the sample was lyophilized overnight for phosphoproteomic analysis.

**Global proteome analysis.** The dried TMT labeled peptide mixture (500 µg) was subjected to high pH reverse phase fractionation on the Agilent 1100 HPLC system (Agilent Technologies, Santa Clara, CA). Peptide mixture was reconstituted in 75 µL of solvent A (5% ACN/50 mM ammonium bicarbonate, pH 8.0) and separated on a Zorbax 300Extend-C18, 3.5 µm, 4.6 × 150 mm column (Agilent Technologies, Santa Clara, CA) at a flow rate of 0.5 mL/min. A gradient from 15%–45% solvent B (90% ACN/50 mM ammonium bicarbonate, pH 8.0) was applied over 49 min with a total run time of 75 min. 96 fractions were collected at 0.63 min interval and every 25th fraction was combined into a set of 24 fractions. Fractions were acidified with 20% TFA, dried, and desalted with SDB tips (GL Sciences, Torrance, CA) prior to mass spectrometry analysis.

**Phosphoproteomic analysis.** Phosphopeptide enrichment was performed using 5 TiO2 cartridges (100 mg absorbent, GL Sciences, Torrance, CA) following the manufacturer's protocol. The following buffers were prepared for this process, buffer A was a mixture of 2% TFA: ACN at 1:4 ratio, buffer B was a mixture of buffer A:lactic acid at 3:1 ratio. Lactic acid was from GL Sciences (Torrance, CA) at 85%–92% concentration. Briefly, the dried peptide mix was reconstituted in buffer B to a concentration of ~ 3 mg/mL and loaded on the TiO2 cartridges that were previously wet and equilibrated with 200 µL of buffers A and B, respectively. The flowthrough was collected and re-loaded onto the cartridges followed by one wash with buffer B, and three washes with buffer A, 200 µL/wash. Phosphopeptides were eluted with 200 µL of 5% NH4OH and 200 µL of 60% ACN/H2O. The equilibration, phosphopeptide binding, and wash steps were performed in the centrifuge at 100 g. The eluents were combined, acidified with 20% TFA, and dried. Phosphopeptides were desalted using SOLA HRP SPE cartridge (10 mg absorbent) from ThermoFisher (San Jose, CA) and dried to completion. The dried phosphopeptide mixture was fractionated as described in global proteome section. The gradient was from 5% to 35% solvent B over 49 min. The fractions from the 96 well plate were pooled such that all 8 fractions/column were pooled into 1 fraction for

a set of 12 fractions. Each fraction was acidified with 20% TFA and dried and desalted with SDB tips (GL Sciences, Torrance, CA).

**Mass spectrometry analysis.** Desalted peptides were reconstituted in 2% ACN/0.1% formic acid (FA)/water and loaded onto Aurora Series 25 cm × 75 µm I.D. column (IonOpticks) using a a Dionex Ultimate 3000 RSLC nano Proflow system (ThermoFisher). Peptide separation was performed at 300 nL/min with a two-step linear gradient where solvent B (0.1% FA/2% water/ACN) was increased from 4% to 30% over 68 min then from 30% to 75% B over 4.9 min with a total analysis time of 95 min. Peptides were analyzed using an Orbitrap Eclipse instrument (ThermoFisher Scientific, San Jose, CA). For global proteome analysis, a real-time search against a human database was employed using an in-house instrument API program called InSeqAPI.1,2,3 Protein-closeout was employed (three distinct peptides/protein/run). The following proteins IRF4, ERN1, XBP1, DGAT2, RB, CCND2, CDK2, UHRF1, CDN1A, CDN1B, SP1, PP2AA, PP2AB, PP2BA, PP2BB, PP2BC, 2AAA, 2AAB, P2R3A, PTPA, 2A5B, 2A5E, 2A5D, 2ABA, 2ABB, 2A5A, 2A5G, 2ABD, 2ABG, P2R3B, STRN, LCMT1, IGBP1, SPY2, STRN3, STRN4, RACK1, and PPME1 were placed on the InSeqAPI inclusion list such that they were not subjected to protein closeout. Precursor ions (MS1) were analyzed in the Orbitrap (250% normalized AGC target, 120,000 mass resolution, 50 ms maximum injection time) with 10 most abundant species were selected for MS2 fragmentation. Each precursor was isolated at a mass width of 0.5 followed by fragmentation using collision-induced dissociation (CID at 30 NCE, 150% normalized AGC target, maximum ion time of 100 ms), and fragment ions were detected in the ion trap. Synchronous precursor selection (SPS) MS3 scans were analyzed in the Orbitrap at 50,000 resolution with the top 8 most intense ions in the MS2 spectrum subjected to HCD fragmentation (45 NCE, AGC target = 3.0E5, maximum injection time of 400 ms). For phosphoproteomic analysis, data was acquired without InSeqAPI real-time search, parameters for MS1 and MS2 scans were the same as above. Synchronous precursor selection MS3 scans were also analyzed in the Orbitrap at 30,000 mass resolution with the top 8 most intense ions in MS2 scan were selected for HCD fragmentation (40% NCE, 250% AGC target, maximum injection time of 300 ms). Data available in the MassIVE repository with the identifier: MSV000095907.

**Bioinformatics.** MS/MS data was searched using the Comet search algorithm against a concatenated forward-reverse target-decoy database (UniProtKBconcat, downloaded Oct 2023) consisting of Homo sapiens proteins and common contaminant sequences. Spectra were assigned using a precursor mass tolerance of 20 ppm and fragment ion settings for low-resolution MS/MS with a fragment ion bin tolerance and bin offset of 1.0005 and 0.4, respectively. Static modifications included carbamidomethyl cysteine (+57.0215 Da), TMT tag (+ 304.2071 Da) on both the N-termini of the peptides and lysine residues. Variable modifications included oxidized methionine (+15.994 Da) and TMT tag on tyrosine residues (+304.2071 Da). For phosphoproteomic analysis, additional variable modification was specified for serine (+79.9663 Da), threonine (+79.9663 Da), and tyrosine (+79.9663 Da). Trypsin specificity with up to 1 and 2 miscleavages was specified for proteome and phosphoproteomic analyses respectively. Peptide spectral matches were filtered at 1% false discovery rate (FDR) for proteome analysis. For phosphoproteomics, peptide spectral matches were filtered at 5% FDR followed by protein filtering at 2% FDR. AScore algorithm was used for phosphorylation site localization. A subset of the data containing phosphopeptides was filtered for statistical analysis. The TMT reporter ion quantification was performed using an in-house Mojave module5 by calculating the highest peak within 20 ppm of theoretical reporter mass windows and correcting for isotope purities.

**Statistical analysis of mass spectrometry data.** The R package MSstatsPTM v.1.2.46 & 7 was used for preprocessing PSM-level quantification, performing global proteome profiling (GPP) data and PTM quantification for PTM data, and conducting differential abundance analysis. Briefly, for both GPP and PTM data, the preprocessing phase filtered out TMT peaks in MS3 scans with a sum of less than 30,000 across all 16 channels. Additionally, peptides shorter than seven residues were eliminated. The abundance of each identified protein or PTM per sample was estimated using Tukey Median Polish summarization, imputing missing values below a censoring threshold of $2^8$. The estimated log2-fold change for each protein or PTM, as well as its associated standard error, was calculated using a linear mixed-effect

model. Subsequently, MSstatsPTM combined these estimated log2-fold changes and their standard errors for adjusting protein-level changes in PTM abundance. For testing the two-sided null hypothesis of no changes in abundance, model-based test statistics were compared to the Student's *t* test distribution with the degrees of freedom appropriate for each protein or PTM. The resulting *P* values were adjusted to control the FDR with the method by Benjamini-Hochberg.

## RNA sequencing

AMO1 shIRE1 RNA-seq and analysis have been described previously [15]. For Bulk RNA sequencing of AMO1 shIRF4 cl.1 or shNTC (non-targeting control), cells were harvested in biologic triplicates at 0, 8, 16, and 24 h post-Doxycycline treatment (0.2 µg/mL). RNA sequencing was performed as described before [10]: Total RNA was extracted using RNeasy Mini Kit (Qiagen) and the quality of the RNA was determined by a Fragment Analyzer (Advanced Analytical Technologies). As input material, 0.1 µg of total RNA was used for library preparation using TruSeq Stranded Total RNA Library Prep Kit (Illumina). Size of the libraries was confirmed using 4200 TapeStation and High Sensitivity D1K screen tape (Agilent Technologies) and RNA concentration was determined by a qPCR-based method using Library quantification kit (KAPA). The libraries were multiplexed and then sequenced on Illumina HiSeq1000 (Illumina) to generate 30M of single-end 50 base-pair reads. RNA-sequencing data were analyzed using HTSeqGenie in BioConductor as follows: first, reads with low nucleotide qualities (70% of bases with quality <23) or matches to rRNA and adapter sequences were removed. The remaining reads were aligned to the human reference genome (human: GRCh38.p10) using GSNAP (PMID:20147302, 27008021) version "2013-10-10-v2", allowing maximum of two mismatches per 75 base sequence (parameters: "-M 2 -n 10 -B 2 -i 1 -N 1 -w 200000 -E 1 --pairmax-rna=200000 --clip-overlap"). Transcript annotation was based on the Gencode genes data base (human: GENCODE 27). To quantify gene expression levels, the number of reads mapping unambiguously to the exons of each gene was calculated. For visualization purposes, count normalization was performed with the formula log2((counts + 1*SF)/(N/10^6). Differentially expressed genes (DEGs) were determined by Partek Flow Explore Spatial Multiomics Data using Partek Flow software, v11.0. with gene differential filter analysis based on FDR < 0.05 and Log2 fold change (FC) > 1.2. Following, gene-set enrichment and leading-edge analysis was conducted with the GSEA_4.3.2 software [70], following the provider's instructions. Parameters used for analysis can be found in S1 Data for the individual figues. Overrepresentation analysis (ORA) was performed using the Interactive Enrichment Analysis Tools developed by the Gladstone Institutes.

## ChIP-sequencing

ChIP-seq was performed as previously described with the following modifications [71]. AMO1 cells (10 × 10⁶/condition) were double crosslinked by 50 mM DSG (disuccinimidyl glutarate, #C1104 - ProteoChem) for 30 min followed by 10 min of 1% formaldehyde. Formaldehyde was quenched by the addition of glycine. Nuclei were isolated with ChIP lysis buffer (1% Triton x-100, 0.1% SDS, 150 mM NaCl, 1 mM EDTA, and 20 mM Tris, pH 8.0). Nuclei were sheared with Covaris sonicator using the following setup: Fill level—10, Duty Cycle—15, PIP—350, Cycles/Burst—200, Time—8 min). Sheared chromatin was immunoprecipitated overnight with the following antibodies: RNAPII-pS2 (Abcam – ab5095) and IRF4 (Cell Signaling – 4964S). Antibody chromatin complexes were pulled down with Protein A magnetic beads and washed once in IP wash buffer I. (1% Triton, 0.1% SDS, 150 mM NaCl, 1 mM EDTA, 20 mM Tris, pH 8.0, and 0.1% NaDOC), twice in IP wash buffer II. (1% Triton, 0.1% SDS, 500 mM NaCl, 1 mM EDTA, 20 mM Tris, pH 8.0, and 0.1% NaDOC), once in IP wash buffer III. (0.25 M LiCl, 0.5% NP-40, 1mM EDTA, 20 mM Tris, pH 8.0, 0.5% NaDOC) and once in TE buffer (10 mM EDTA and 200 mM Tris, pH 8.0). DNA was eluted from the beads by vigorous shaking for 20 min in elution buffer (100 mM NaHCO₃, 1% SDS). DNA was de-crosslinked overnight at 65C and purified with MinElute PCR purification kit (Qiagen). DNA was quantified by Qubit and 10 ng DNA was used for sequencing library construction with the Ovation Ultralow Library System V2 (Tecan) using 14 PCR cycles according to the manufacturer's recommendations. Libraries were sequenced with Illumina Nextseq, using paired-end 50bp read configuration.

## ChIP-seq analysis

ChIP-seq results were analyzed using the ENCODE ChIP-seq pipeline (v2.2.1) [72]. ChIP-seq reads were aligned to the human reference genome (hg38) using Bowtie2 (v2.3.4.3) [73]. Aligned reads were then filtered for quality and duplicates using samtools (v1.9) [74] and Picard (Broad Institute - v2.20.7). The SPP peak caller was used to call ChIP-seq peaks, and input was used to assess the background of the experiments [75]. Peak sets were filtered using a list of genomic regions that contain anomalous, unstructured, or experiment-independent high signal [76]. ChIP-seq bam and bed files were then used to call differential peaks in the genome by using DiffBind (v3.12.0) [77]. The rest of the analyses were conducted using a combination of "DiffBind", "edgeR" [78], HOMER [49], and deepTools2 [79]. Briefly, DiffBind was employed for differential binding analysis. Read counts were normalized using depth normalization. The "edgeR" method was applied for statistical analysis, using a FDR threshold of <0.05 to identify differentially bound regions. HOMER was then used to perform motif enrichment analysis on these BED files, with motif lengths ranging from 6 to 16 bp, using the hg38 genome as a reference. Coverage plots were generated using deepTools to visualize ChIP-seq signal across the identified genomic regions. Bigwig files for IRF4 and RNAPIIpS2 were processed, and coverage matrices were computed with "computeMatrix". Heatmaps and profile plots were generated using "plotHeatmap" and "plotProfile", respectively. This comprehensive approach allowed for the detailed characterization of differential binding events and their potential regulatory motifs in the IRF4 and RNAPII datasets. Finally, JASPAR was used to identify transcription factor motifs in specific regulatory regions [53].

## Cell cycle analysis (PI staining)

$1 \times 10^6$ cells were fixed in ice-cold 70% ethanol overnight at 4°C. After washing in PBS, samples were treated with 100 µg/mL RNase (Zymo research) for 15 min, RT, followed by incubation with 50 µg/mL PI for 20 min at RT. Samples were analyzed using a FACSymphony analyzer (BD) and data were collected in linear scale for better DNA content determination. Univariate modeling (either Watson pragmatic or Dean Jett fox) was used to create a fit to cell cycle data based on statistics in the DNA content dimension and was performed with the FlowJo 10.10.0 software "cell cycle" function, constraining the model to equal G2 and G1 CVs.

## Statistical analysis

All values represent the mean ± standard error of at least three biological replicates. Statistical analysis of the results was performed by unpaired, two-tailed *t* test or ANOVA followed by an appropriate post-hoc analysis, including Bonferroni correction to compensate for multiple comparisons. All statistical analyses were performed with GraphPad Prism 10 (GraphPad Software, Inc.). *P*-Values above 0.05 were considered as not significant.

## Schematics

All schematics were created with BioRender.com.

## Supporting information

**S1 Fig. Silencing of IRE1 but not XBP1 downregulates IRF4.** (A) *IRF4* mRNA abundance in IRE1 or XBP1-depleted L363 cells. L363 shIRE1 Cl.C versus L363 shXBP1 Cl.N cells were treated with Dox (0.2 µg/mL) for the indicated time and analyzed by RT-qPCR for levels of *ERN1*, *IRF4* and *XBP1s*. Data represented as mean ± SEM. **(B)** IRF4 protein abundance in IRE1 or XBP1-depleted L363 cells. L363 shIRE1 Cl.C or L363 shIRE1 Cl.D versus L363 shXBP1 Cl.N cells were treated with Dox (0.2 µg/mL) for the indicated time and were analyzed by IB for IRF4 protein levels. **(C)** *Corrected* gene effect scores of IRE1 codependencies, in KMS11: Chronos gene-effect values from KMS11 genes (derived from Depmap) were "corrected" to reveal MM-specific dependencies. Annotated are genes decreased 2-fold or more by IRE1 silencing

in AMO1 cells (red in Fig 1A). **(D)** IRF4 protein abundance in IRE1 or XBP1-depleted KMS11 cells. KMS11 shIRE1 or KMS11 shXBP1 Cl.8 cells were treated with Dox (0.2 µg/mL) for the indicated time and analyzed by IB for IRF4 protein levels. **(E)** Effect of IRE1 silencing or inhibition on *in vitro* spheroid growth of H929 cells. Cells were stably transfected with plasmids encoding Dox-inducible shRNAs against either IRE1 (red) or non-targeting control (blue). Growth of these cells in the absence (closed symbols) or presence (open symbols) of Dox (0.2 µg/mL) was compared to that of cells expressing shNTC or cells treated with IRE1 RNase (RI; 1 µM) or kinase (KI; 1 µM) inhibitors. Spheroid growth, depicted as FC confluence, was monitored by time-lapse microscopy in an IncuCyte instrument and values represent mean ± SEM. **(F)** IRF4 protein abundance in IRE1-depleted H929 cells. H929 cells stably transfected with Dox-inducible shRNAs against IRE1 or non targeting control (NTC) were treated with Dox (0.2 µg/mL) for the indicated time and analyzed by IB for IRF4 protein levels. **(G)** Effect of IRE1 silencing on *in vitro* spheroid growth in IRE1-independent T-cell lymphoma, SR796 cells. Cells were stably transfected with plasmids encoding Dox-inducible shRNA against either IRE1. Growth of these mock-treated cells (closed symbols) was compared to that of cells treated with Dox (0.2 µg/mL) or IRE1 inhibitors (RI or KI; 1 µM). Spheroid growth, depicted as FC confluence, was monitored by time-lapse microscopy in an IncuCyte instrument and values represent mean ± SEM. **(H)** IRF4 protein abundance in SR786. SR786 shIRE1 Cl. 7 cells were treated with Dox (0.2 µg/mL) for the indicated time and analyzed by IB for IRF4 protein levels. **(I)** Schematic representation of how IRF4 and IRE1 dependencies intertwine. All cell lines are IRF4 dependent while IRE1 enzymatically dependent cell lines are depicted in blue and nonenzymatically dependent cell lines are depicted in red. **(J),(K)** Validation of IRE1 inhibition based on XBP1s and RIDD-target mRNA. Samples from Fig 1G–J were analyzed by RT-qPCR for *XBP1s, DGAT2* and *CD59* (RIDD targets). Data represented as mean ± SEM. The data underlying this figure can be found in S1 Data and S1 Raw images.
(TIFF)

**S2 Fig. IRE1 silencing attenuates IRF4 activity.** **(A)** Effect of IRE1 silencing on *IRF4* mRNA stability. AMO1 shIRE1 Cl.1 (left) or shXBP1 Cl.1 (right) cells were pre-treated with or without 0.2 µg/mL Dox for 48 h and then 1 µg/mL Actinomycin D (ActD) was added for 3 h or 6 h. Samples were analyzed by RT-qPCR for *ERN1*, *IRF4* and *XBP1s*. Data represented as mean ± SEM. **(B)** Effect of IRE1 silencing on Myc protein levels in KMS27. KMS27 shIRE1 cells underwent a Dox (0.2 µg/mL) time course and were analyzed by IB for IRF4 and Myc. **(C)** Effect of IRE1 silencing on IRF4-regulating transcription factor chromatin-binding activity. AMO1 shIRE1 Cl.1 cells were cultured in the absence or presence of 0.2 µg/mL Dox for 72 h. Cells were sequentially lysed into 4 subcellular fractions: C – cytoplasmic, M – Membrane, SN – Soluble Nuclear, CN – Chromatin-bound Nuclear. Nuclear fractions were analyzed by IB for IRE1, MITF, Myc, and Aiolos while Cofilin, Histone H3, and Lamin B2 serve as fractionation controls. *Right:* The ratio of chromatin-bound over soluble nuclear MITF, Aiolos, and Myc was determined by densitometry and is depicted relative to mock-treated cells. **(D)** Validation of MYC protein depletion by shMYC silencing. AMO1 samples from Fig 2E were analyzed by IB for MYC, IRE1, and IRF4. GAPDH used as a loading control. **(E)** Validation of WT and phospho-mutant IRF4 ectopic expression. AMO1 WT IRF4 versus AMO1 S114A_S270A versus S114D_270D were treated in the absence or presence of 0.2 µg/mL Dox for 72 h. The samples were analyzed by IB for IRF4 protein since the mutant IRF4 constructs are not recognized by complementary to endogenous IRF4 PCR primers. **(F)** Substitution of S114 and S270 by alanine versus aspartic acid and its effect on IRF4's chromatin-binding activity. Samples from Fig 2H were analyzed by IB as in Fig 2F: C – cytoplasmic, M – Membrane, SN – Soluble Nuclear, CN – Chromatin-bound Nuclear and were analyzed by IB for IRF4 before this was quantified and normalized for Fig 2H. The data underlying this figure can be found in S1 Data and S1 Raw images.
(TIFF)

**S3 Fig. IRF4 silencing recapitulates the anti-proliferative effect of IRE1 knockdown.** **(A)** Validation of IRF4 depletion by IRF4 silencing. AMO1 shIRE1 Cl.1 or shIRF4 Cl.1 cells were stably transfected with plasmids encoding Dox-inducible

shRNAs against either IRF4 or IRE1. The cells were treated with Dox (0.2 µg/mL) for the indicated time. Samples were analyzed by IB for IRE1 and IRF4. **(B)** Effect of IRF4, IRE1 or NTC silencing on *in vitro* spheroid growth of KMS27 and validation of IRF4 silencing. KMS27 cells were stably transfected with plasmids encoding Dox-inducible shRNAs against either IRF4 (purple) or non-targeting control (blue). *Left, middle:* KMS27 shNTC, shIRE1, or shIRF4 cells were treated with Dox (0.2 µg/mL) for the indicated time. Samples were analyzed by RT-qPCR and IB for IRE1/*ERN1* and IRF4. *Right:* Growth of these cells in the absence (closed symbols) or presence (open symbols) of Dox (0.2 µg/mL) was compared to that of cells expressing shRNAs against IRE1 or NTC. Spheroid growth, depicted as FC confluence, was monitored by time-lapse microscopy in an IncuCyte instrument and values represent mean ± SEM. **(C)** Cell death markers. AMO1 shIRE1 Cl.1, shIRF4 Cl.1, or shXBP1 Cl.1 cells were treated in the absence or presence of Dox (0.2 µg/mL) for up to 72 h and post-nuclear lysates were analyzed by IB for Lamin A/C and PARP1 cleavage as well as Histones. GAPDH is used as a loading control. **(D)** Effect of IRE1 or IRF4 silencing on caspase activation. AMO1 shIRE1 Cl.1 or shIRF4 Cl.1 cells treated in the absence (filled bars) or presence of Dox (0.2 µg/mL) for 72 h were analyzed for caspase activity by Caspase-Glo assays. Representative replicate. Values presented as mean ±SEM. **(E)** Effect of IRE1 or IRF4 silencing on Annexin V/ PI staining. AMO1 shIRE1 Cl.1 or shIRF4 Cl.1 cells were treated with Dox (0.2 µg/mL) for the indicated times. Cells were then stained with FITC-Annexin V and PI and analyzed by flow cytometry for early apoptotic (FITC+ PI-), late apoptotic (FITC+ PI+), and necrotic (FITC- PI+) cells. **(F)** Q-VD blockade of caspase cleavage during IRE1 or IRF4 silencing. AMO1 shIRE1 Cl.1 or shIRF4 Cl.1 cells were treated with Dox (0.2 µg/mL) for 72 h in the absence or presence of the pan-caspase inhibitor Q-VD (50 µM). Samples were analyzed by IB for caspase and PARP1 cleavage. **(G)** Viability of IRE1 or IRF4-deficient cells during Q-VD treatment. Cells treated as in S3F Fig were analyzed for plasma-membrane integrity/viability by trypan-blue staining and quantified in a Vi-Cell machine. **(H)** *In vitro* spheroid growth of IRF4 or IRE1 deficient cells during Q-VD treatment. AMO1 shIRE1 Cl.1 or shIRF4 Cl.1 cells were cultured in the absence (filled symbols) or presence (open symbols) of Dox (0.2 µg/mL) with or without Q-VD (50 µM) and spheroid cell growth, depicted as FC confluence, was monitored by time-lapse microscopy in an IncuCyte instrument. Values presented as mean ±SEM. **(I)** Validation of caspase inactivation. Samples from S3H Fig were analyzed for caspase activity by Caspase 3/7-Glo assay. Values presented as mean ±SEM. **(J)** Effect of IRE1 and IRF4 silencing on cell divisions in KMS27 cells. KMS27 shIRE1, shIRF4, or shNTC were stained with a CFSE-type dye and incubated in the absence (filled curves) or presence (open curves) of Dox (0.2 µg/mL) and analyzed by flow cytometry. Etoposide (Eto, 25 µM, dashed line) was used as a non-proliferative control. Representative experiment. **(K)** Control panels for Fig 3C. Quantification of apoptotic (left) versus dividing (right) cells. **(L)** Cell cycle profiling in KMS27 cells. KMS27 shIRF4 cells were incubated in the absence (filled symbols) or presence (open symbols) of Dox (0.2 µg/mL) for the indicated times, EtOH fixated and PI stained before analyzed by flow cytometry. The indicated cell cycle phases were determined according to univariate (DNA content) modeling. **(M)** Effect of Myc silencing on in vitro spheroid growth and cell cycle in AMO1 cells. Left: Growth of AMO1 shMYC Cl. 7 (yellow) in the absence of any treatment (closed symbols) was compared to that of cells treated with Dox (0.2 µg/mL) as well as to the growth of AMO1 shIRE1, shIRF4, and shNTC cells in the presence of Dox. Spheroid growth, depicted as FC confluence, was monitored by time-lapse microscopy in an IncuCyte instrument and values represent mean ± SEM. *Right:* G1-phase profiling of the same cell lines. Cells were incubated in the presence of absence of Dox for 48 h, EtOH fixated and PI stained before analyzed by flow cytometry. G1 cell cycle phase was determined according to univariate (DNA content) modeling. Values represented as mean ± SEM. **(N)** CDK2 phosphorylations during IRE1 or IRF4 silencing in KMS27 cells. KMS27 shIRE1 Cl.9, shIRF4 Cl.20, or shXBP1 Cl.22 cells were treated in the presence or absence of Dox (0.2 µg/mL) in a time course of 0, 6, 24, 48, and 72 h and analyzed by IB for total CDK2 and pT160 CDK2. **(O)** IRF4 silencing effect on inhibitory and activating CDK phosphorylations. AMO1 shIRF4 Cl.1 cells were cultured in the absence (filled symbols) or presence (open symbols) of Dox (0.2 µg/mL) and collected at the indicated time post-treatment to be analyzed by IB for total CDK6, pT24 CDK6 (inhibitory), total CDK2, pT14 CDK2 (inhibitory), and pT160 CDK2 (activating). Bottom panels: quantification by densitometry. **(P)** Cdc25A protein expression during IRF4 silencing. AMO1 shIRF4 Cl.1 or

shNTC cells were treated in the absence or presence of Dox (0.2 μg/mL) for the indicated time. Cells were analyzed by IB for total or phosphoT124-Cdc25A as well as total CDK levels. Right: quantification of Cdc25A by densitometry in shNTC (filled symbols) or shIRF4 (open symbols) cells. The data underlying this figure can be found in S1 Data, **Zenodo (**https://doi.org/10.5281/zenodo.14928364**), and** S1 Raw images.
(TIFF)

**S4 Fig.  IRE1 and IRF4 regulate a highly overlapping set of cell cycle genes.** (A) Validation of samples from Fig 4. AMO1 shIRF4 Cl.1 or shNTC bulk RNA-seq analysis (addressed in Fig 4), validates that IRF4 transcripts were depleted during the time course of silencing, as well as transcripts of known IRF4 targets. i.e., *MYC*, *PRDM1* (transcript of BLIMP1) and *XBP1*. **(B)** Complete IRF4 GSEA (Hallmark). GSEA analysis was performed as described in Fig 4A. Shown all enriched datasets. In gray, gene sets with FDR > 0.02. **(C)** ORA of AMO1 shIRE1 (top) and shIRF4 (bottom). Overrepresentation analysis of the transcriptomics results was performed using the Interactive analysis enrichment tool. Depicted are select GO terms. **(D)** Overlap of genes between IRE1 and IRF4 Knockdowns. After GSEA in Fig 4A, Leading Edge Genes (S2 Table), representing those contributing most significantly to the enrichment of the given gene sets, were extracted for both genetic backgrounds. Shown: Venn diagrams illustrating the intersection between the Leading-Edge Genes for "G1S Cell Cycle Control" (Wiki Pathways) and "Unfolded Protein Response" (UPR; Hallmark) between IRE1 knockdown (shIRE1) and IRF4 knockdown (shIRF4) backgrounds. Middle: Venn diagram representing all DNA repair GO term genes identified in the two backgrounds and their overlap. **(E)** UPR downregulation was validated by IB in both genetic backgrounds. Samples treated as in Fig 4D were analyzed by IB for pIRE1 as well as PERK and ATF6 pathway proteins. **(F)** Effect of IRE1 or IRF4 knockdown on mRNA expression of DNA repair genes. Heatmap depicting the top 100 downregulated genes match to "DNA repair" GO term in the transcriptomics analyses described in Fig 4A. **(G)** Complete IRF4 GSEA (Hallmark) analysis in KMS27 cells. Analysis was performed as described in Fig 4A for KMS27 cells. Shown all enriched datasets. In gray, gene sets with FDR > 0.02. *Right:* Overlap of Leading-Edge Genes from GSEA analyses in IRF4-deficient AMO1 and KMS27 cells. After GSEA in Fig 4A, Leading Edge Genes (S2 Table), representing those contributing most significantly to the enrichment of the given gene sets, were extracted for both cell lines. Shown: Venn diagrams illustrating the intersection between the Leading-Edge Genes for Hallmark "E2F-Targets" and "G2/M Checkpoint" between KMS27 and AMO1 cells deficient in IRF4. The data underlying this figure can be found in S1 Data, S1 Raw images, and GEO repository (GSE288674).
(TIFF)

**S5 Fig.  IRF4 directly controls E2F1 gene transcription, which sufficiently regulates cell cycle downstream of IRF4.** (A) IRF4 differential binding within the PRDM1 locus—a characterized IRF4 target—by ChIP-seq differential binding analysis. Genome browser tracks depicting the intensity of IRF4 (black and green) or RNAPIIpS2 (red and blue) binding in IRF4 proficient or deficient cells, respectively. Yellow shadowing indicates the differential binding regions identified by the analysis in Fig 5A. **(B)** IRF4 differential binding within the *IRF4* locus by ChIP-seq differential binding analysis. Genome browser tracks depicting the intensity of IRF4 (black and green) or RNAPIIpS2 (red and blue) binding in IRF4 proficient or deficient cells, respectively. Yellow shadowing indicates the differential binding regions identified by the analysis in Fig 5A. **(C)** Genome browser depiction of ChIP-seq signals across *RB1* locus. Genome browser tracks depicting the intensity of IRF4 (black and green) or RNAPIIpS2 (red and blue) binding in IRF4 proficient or deficient cells, respectively. No differential binding regions were identified by differential binding analysis. **(D)** IRF4 differential binding within the *CDC25A* locus by ChIP-seq differential binding analysis. Genome browser tracks depicting the intensity of IRF4 (black and green) or RNAPIIpS2 (red and blue) binding in IRF4 proficient or deficient cells, respectively. Yellow shadowing indicates the differential binding region identified by the analysis in Fig 5A. **(E)** IRF4 differential binding within the *UHRF1* locus by ChIP-seq differential binding analysis. Genome browser tracks depicting the intensity of IRF4 (black and green) or RNAPIIpS2 (red and blue) binding in IRF4 proficient or deficient cells, respectively. Yellow shadowing indicates the differential binding

region identified by the analysis in Fig 5A. The data underlying this figure can be found in GEO repository (GSE288671) and S2 Data.
(TIFF)

**S6 Fig. IRF4 repletion in IRE1-deficient cells rescues E2F1 expression and proliferation.** (A) Effect of IRF4 expression levels in the growth of IRF4 rescued cell lines. AMO1 shIRF4 cells were stably transfected with Dox-inducible IRF4 or EV. AMO1 shIRF4 EV or individual shIRF4 IRF4 clones were cultured in the absence (closed symbols) or presence (open symbols) of Dox (0.1 µg/mL). *Left:* Spheroid growth, depicted as FC confluence, was monitored by time-lapse microscopy in an IncuCyte instrument and values represent mean ± SEM. *Right:* Analysis of these cells after 24 h of culture in the same conditions by RT-qPCR for *IRF4* mRNA levels. Dashed line represents the level of endogenous IRF4 expression. Values represent mean ± SEM. **(B)** Effect of ectopic IRF4 expression in the growth of AMO1 cells. AMO1 shNTC cells were stably transfected with inducible IRF4 or EV. The cells were then cultured in the absence (closed symbols) or presence (open symbols) of Dox (0.1 µg/mL) and spheroid growth, depicted as FC confluence, was monitored by time-lapse microscopy in an IncuCyte instrument and values represent mean ± SEM. *Right:* Analysis of these cells after 24 h of culture in the same conditions by RT-qPCR for *IRF4* mRNA levels. Values represent mean ± SEM. **(C)** Caspase activation upon IRF4 re-expression in IRE1-deficient cells. AMO1 shIRE1 Cl.1 EV versus shIRE1 Cl.1 IRF4 Cl.4 cells were treated in the absence (filled symbols) or presence (open symbols) of Dox (0.2 µg/mL) for the indicated times were analyzed for caspase activity by Caspase 3/7-Glo assay. Values presented as mean ±SEM. **(D)** Effect of IRE1 silencing on DR5 levels compared to the effect of ectopic IRF4 expression in IRE1 deficient cells. AMO1 shIRE1 EV or IRF4 Cl.4 as well as AMO1 shIRF4 EV or IRF4 Cl.5 were incubated in the absence or presence of 0.1 µg/mL Dox for 72 h. The samples were then analyzed for DR5 levels by IB. Cleaved Caspases served as apoptosis markers. **(E)** Effect of IRF4 repletion on E2Fs and other targets during IRF4 silencing. AMO1 shIRF4 Cl.1 EV or shIRF4 Cl.1 IRF4 Cl. 5 cells were cultured in the absence or presence of Dox (0.1 µg/mL) for 24 h and analyzed by RT-qPCR for *E2F1* and *UHRF1* mRNAs. **(F)** Effect of IRF4 repletion on E2Fs and other targets' protein levels during IRF4 silencing. AMO1 shIRF4 Cl.1 EV or shIRF4 Cl.1 IRF4 Cl. 5 cells were cultured in the presence of Dox for the indicated times and analyzed by IB for E2F1, UHRF1, CDC25A, CDK2 and p21. **(G)** Ability of ectopic WT or phospho-mutant IRF4 for chromatin binding in the absence of endogenous IRF4. AMO1 shIRE1 Cl.1 cells expressing Dox-inducible WT IRF4, S114A_S270A IRF4 or S114D_S270D IRF4 were treated with or without 0.2 µg/mL Dox for 72 h. Cells were sequentially lysed into 4 subcellular fractions: C – cytoplasmic, M – Membrane, SN – Soluble Nuclear, CN – Chromatin-bound Nuclear. Nuclear fractions were analyzed by IB for IRE1, IRF4, and c-Myc while Cofilin, Histone H3 and Lamin B2 serve as fractionation controls. **(H)** Quantification of IRF4 and Myc chromatin binding after IRF4 ectopic expression in cells harboring IRE1 depletion. The IBs from S6G Fig were quantified by densitometry. The ratio of chromatin-bound over soluble nuclear IRF4 and Myc was first calculated relative to mock-treated cells and then depicted as a fold change to the cell line expressing WT IRF4. **(I)** Effect of WT versus phosphor-mutant IRF4 repletion on nascent *PRDM1* transcription. Nascent RNA from Fig 6J samples were also analyzed for *PRDM1* by RT-qPCR. Dashed line represents the level of endogenous nascent *PRDM1* expression (mock sample). **(J)** Effect of WT versus phospho-mutant IRF4 repletion on transcriptional levels of *IRF4*, *E2F1* and other targets. AMO1 shIRE1 Cl.1 EV or shIRE1 Cl.1 (WT) IRF4 Cl. 4 or shIRE1 Cl.1 S114A_S270A IRF4 Cl. 1 and shIRE1 Cl.1 S114D_S270D IRF4 Cl. 1 cells were cultured in the absence or presence of Dox (0.2 µg/mL) for 24 h and analyzed by RT-qPCR for *E2F1* mRNAs and E2F1 targets, *UHRF1* and *CDC25A*. The data underlying this figure can be found in S1 Data and S1 Raw images.
(TIFF)

**S7 Fig. Hypothetical model for IRF4 regulation in MM via IRE1 versus other factors.** Schematic representation of IRE1 supporting IRF4 activity/expression in IRE1-dependent MM lines. Two newly identified modes of IRF4 regulation stemming from IRE1 require either XBP1s (enzymatic IRE1 dependency) or an unknown phosphotransferase intermediary that acts

independently of IRE1 enzymatic activity (nonenzymatic dependency). IRE1-independent cell lines are dependent on IRF4 but IRF4 is not regulated by IRE1 but rather by previously identified transcription factors. The latter axes of IRF4 regulation may be present in IRE1-dependent cell lines as well. Created in BioRender. Oikonomidi, I. (2025) https://BioRender.com/b50x169. (TIFF)

**S1 Raw images.  Original raw immunoblot images.** Each blot is labeled to annotate sample order and antibody of blotting. Lanes not included in the final figures are marked with "X" above the lane label. (PDF)

**S1 Data.  Underlying numerical data.** Each sheet of the file represents a figure panel and includes all values used to generate the respective graphs in the figures. Some of these datasets are complemented by RNA-seq, ChIP-seq or proteomics data that have been submitted to GEO or MassIVE (accession numbers provided in the corresponding tabs), respectively, and are publicly available. (XLSX)

**S2 Data.  Underlying numerical data for** Fig 5A**.** Matrixes were generated by the ChIP-sequencing differential binding analysis described in Materials and Methods. For raw data files as well as bigwig files, please refer to GEO repository with accession number GSE288671. (ZIP)

**S1 Table.  List of 131 genes comprising nonenzymatic IRE1 co-dependencies and their corrected scores.** Chronos gene effect values derived from Depmap were "corrected" as described in *Methods* to reveal MM-specific dependencies. Subsequently, nonenzymatic-IRE1 co-dependencies were obtained by filtering for those genes with *corrected* score ≤ −0.25 in all four cell lines (AMO1, KMS27, L363, JJN3). (XLSX)

**S2 Table.  Leading edge analysis and overlap.** Data referring to the transcriptomics GSEA analysis of Fig 4. Provided are lists of all the core enrichment genes (leading edge) that were identified in each gene set. These genes were further used for overlap analysis, depicted with the Venn Diagrams in Figs 4B, S4D, and S4G. (XLSX)

## Acknowledgments

We thank Tao Sun for plasmid design, Rachana Pradhan, Ximo Pechuan-Jorge, Anatoly Belov, and Meena Choi for bioinformatic analyses, Yuxin Liang and Manching Ku for coordinating bulk RNA-sequencing, and Steffan Vartanian for methodological guidance regarding CDK2. We thank the Gladstone Bioinformatics Core for their interactive enrichment analysis tools.

## Author contributions

**Conceptualization:** Ioanna Oikonomidi, Avi Ashkenazi.

**Data curation:** Ioanna Oikonomidi, Victoria C. Pham, Lauren M. Gutgesell, Avi Ashkenazi.

**Formal analysis:** Ioanna Oikonomidi, Victoria C. Pham, Bence Daniel.

**Funding acquisition:** Avi Ashkenazi.

**Investigation:** Ioanna Oikonomidi, Vasumathi Kameswaran, Victoria C. Pham, Iratxe Zuazo-Gaztelu, Lauren M. Gutgesell, Bence Daniel.

**Methodology:** Ioanna Oikonomidi, Vasumathi Kameswaran, Victoria C. Pham, Scot Marsters, Bence Daniel.

**Project administration:** Avi Ashkenazi.

**Resources:** Scot Marsters, Avi Ashkenazi.

**Software:** Ioanna Oikonomidi.

**Supervision:** Ioanna Oikonomidi, Bence Daniel, Jennie R. Lill, Zora Modrusan, Avi Ashkenazi.

**Validation:** Ioanna Oikonomidi.

**Visualization:** Ioanna Oikonomidi, Bence Daniel.

**Writing – original draft:** Ioanna Oikonomidi.

**Writing – review & editing:** Ioanna Oikonomidi, Bence Daniel, Avi Ashkenazi.

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
