## [Editor Report · Decision Letter 0]

18 Sep 2024

Dear Dr Ashkenazi, 

Thank you for submitting your manuscript entitled "IRF4 mediates nonenzymatic dependency on IRE1 in Multiple Myeloma cells" for consideration as a Research Article by PLOS Biology.

Your manuscript has now been evaluated by the PLOS Biology editorial staff as well as by an academic editor with relevant expertise and I am writing to let you know that we would like to send your submission out for external peer review.

Once your full submission is complete, your paper will undergo a series of checks in preparation for peer review. After your manuscript has passed the checks it will be sent out for review. To provide the metadata for your submission, please Login to Editorial Manager (https://www.editorialmanager.com/pbiology) within two working days, i.e. by Sep 20 2024 11:59PM.

Kind regards,

Ines

--

Ines Alvarez-Garcia, PhD

Senior Editor

PLOS Biology

---

## [Decision Letter · Decision Letter 1]

16 Nov 2024

Dear Dr Ashkenazi,

Thank you for your patience while your manuscript entitled "IRF4 mediates nonenzymatic dependency on IRE1 in Multiple Myeloma cells" was peer-reviewed at PLOS Biology. It has now been evaluated by the PLOS Biology editors, an Academic Editor with relevant expertise, and by four independent reviewers. 

The reviews are attached below. You will see that the reviewers find the conclusions novel and interesting, however they also think that more experiments are required to confirm the results and to obtain further mechanistic insights on how IRE1 works independently of its enzymatic functions and how critical is IRE1 for myeloma survival. Reviewer 1 wonders whether/how the relationship between IRE1 and IRF4 differs between MM cells that depend on IRE1 vs those that are not dependent on IRE1 activity (but dependent on the protein). Reviewers 2 and 3 think that additional experiments should be performed to confirm that IRE1 nonenzymatic activity is central to the proposed mechanism. These reviewers also suggest other experiments that include rescuing or exploring the role of miRNAs, confirming that IRE1 has a role in regulating nascent IRF4 mRNA and excluding the role of other transcription factors beyond their own transcriptional levels in the presence of IRE1 knockdown. In addition, Reviewer 2 thinks the role of IRF4 phosphorylation in sustaining the generation of native IRF4 mRNA via the IRE1-driven chromatin binding of IRF4 should be clearly defined. Reviewer 4 asks for clarifications on the methods and other points that are important for understanding the conclusions, but also suggests an experiment to explore what happens to IRF4 when IRE1 is silenced in MM cell lines that are not dependent on IRE1 for their survival.

In light of the reviews, which you will find at the end of this email, we would like to invite you to revise the work to thoroughly address the reviewers' reports. Given the extent of revision needed, we cannot make a decision about publication until we have seen the revised manuscript and your response to the reviewers' comments. Your revised manuscript is likely to be sent for further evaluation by all or a subset of the reviewers.

**IMPORTANT - SUBMITTING YOUR REVISION**

3. Resubmission Checklist

a) *PLOS Data Policy*

b) *Published Peer Review*

d) *Blurb*

Please also provide a blurb which (if accepted) will be included in our weekly and monthly Electronic Table of Contents, sent out to readers of PLOS Biology, and may be used to promote your article in social media. The blurb should be about 30-40 words long and is subject to editorial changes. It should, without exaggeration, entice people to read your manuscript. It should not be redundant with the title and should not contain acronyms or abbreviations. For examples, view our author guidelines: https://journals.plos.org/plosbiology/s/revising-your-manuscript#loc-blurb

Sincerely,

Ines

--

Ines Alvarez-Garcia, PhD

Senior Editor

PLOS Biology

Reviewers' comments

Rev. 1:

This study is a follow-up to a parallel study (which I also reviewed) showing that a subset of multiple myeloma (MM) cells are dependent on IRE1 protein but not IRE1 enzymatic activity. The current work attempts to elucidate factors downstream of this non-enzymatic role of IRE1. Using a combination of transcriptomics and available data from DepMap, the authors identify IRF4 as one of the most consistently downregulated mRNAs in Ire-dependent MM cells. This is validated at the protein level, is shown to not be dependent on Xbp1s (and hence, is related to the non-enzymatic role of IRE1), and seems to mostly be at the level of transcription, with a partial contribution from increased mRNA decay. Further molecular analysis indicates that IRE1 somehow maintains IRF4 in a de-phosphorylated state, which allows it to both self-sustain its levels and act on downstream transcriptional targets that regulate cell cycle progression. Importantly, two lines of evidence are provided that IRE1 is upstream of IRF4: (i) the set of genes affected almost completely overlaps in IRE1-lacking and IRF4-lacking cells; (ii) forced re-expression of IRF4 in IRE1-lacking MM cells rescues many of the key cell cycle consequences.

Overall, this study seems to me to be a major advance in two ways. First, it helps solidify the earlier conclusion that there is a non-enzymatic role for IRE1 in a subset of MM by now identifying a molecular player downstream of this unexplored pathway. Second, it highlights a key role for IRF4 in this subset of MM and provides a rich dataset of downstream consequences in this newly described IRE1-IRF4 axis. I am therefore supportive of publication. It is worth noting that signalling is not my area, and I cannot really evaluate the bioinformatics used in Fig. 1, but the robustness and quality of the data seem to be very high.

Only a couple of minor comments or suggestions:

1) The most important thing I was wondering throughout the manuscript was whether or how the relationship between IRE1 and IRF4 differs between MM cells that are dependent on IRE1 activity versus those that are not dependent on IRE1 activity (but nonetheless dependent on IRE1 protein). Can the authors comment on this? It could be useful in Fig. 1B to perhaps make the same plot for other MM cell lines where IRE1 activity is important. In short, I am wondering whether the authors think this alternative IRE1-IRF4 axis is only operating in the activity-independent MM cells, or is just more evident in these cells because the activity-dependent cells are confounded by other effects of IRE1.

2) Some of the dot plots (e.g., Fig. 4E) and other graphs are very small and hard to read. Perhaps consider harmonizing the size of text and labels across a figure.

Rev. 2: Giovanni Tonon – note that this reviewer has signed his review.

In this manuscript, the Authors aim to define a mechanistic connection between the non-enzymatic activity of IRE-1 and IRF4, both central conduits in the hematolological cancer multiple myeloma (MM). The Authors argue that IRE1 is one of the main drivers of IRF4 activity, specifically on IRF4 regulation of the cell cycle. Much of the data on which the hypothesis of this manuscript rest is based on another manuscript, under consideration and not included. Nonetheless, from what it is possible to gather the results are interesting, and indeed provide useful mechanistic insights on the role of these two central players in MM, and their mutual interactions.

Major points to be addressed:

1. the concept of nonenzymatic activity of IRE1 is central to this manuscript, yet it rests only on suppl fig. 1G-J, where an inhibitor has been used. To conclusively demonstrate this hypothesis, the Authors should reintroduce in cells knocked down for IRE1 either an shRNA-resistant wild type type or inactive IRE1.

2. the role of IRE1 in regulating not simply transcription, but nascent IRF4 mRNA is interesting, however only marginally addressed. It is unclear for example how the assessment of other transcription factors may affect this pattern. More thorough experiments, including rescue experiments, are required to both demonstrate a role for IRE1 in controlling nascent IRF4 mRNA, and to exclude the role of other transcription factors beyond their own transcriptional levels in the presence of IRE1 knock-down (as it was partially performed for MYC). For instance, the putative role of miRNAs, as suggested in the discussion, should be explored.

3. along these lines, it would be important to define the role of IRF4 phosphorylations in sustaining the generation of native IRF4 mRNA, through the IRE1-driven chromatin binding of IRF4.

4. It shoud be explained how the Ser114 as well as Ser270 on IRF4 have been selected.

5. it remains unclear why IRE1 and IRF4 knock-down triggers DNA damage. Could the Authors elaborate on this aspect?

6. how the Authors do intrepret the apparent lack of impact of knock-down of either IRE1 or IRF4 on the G2/M phase, in figure 3D? this finding seems at odds with the effect on G1 and S.

7. it is unclear the impact of IRF4 knock down on the ChIP-seq presented in figure 5A, as the profiles seem overlapping.

Minor points:

- the dots in figure 3A are fairly difficult to read, please replace them with different colors.

Rev. 3:

This is an interesting, but underdeveloped manuscript that reports that some multiple myeloma cell lines require the expression of the endoplasmic reticulum IRE1alpha kinase/RNase but neither of its enzymatic functions.

The IRE1alpha/XBP1 pathway is known to be critical for normal plasma cell development. This pathway is often upregulated in myeloma and some studies (although not all) have shown that genetically or pharmacologically inhibiting this pathway blunts the growth of some myeloma cell lines.

Here, the authors make the interesting observation that a subset of myeloma cell lines fail to grow (and eventually die) when IRE1alpha is genetically silenced but not when XBP1 is genetically silenced. Hence, they appear to have a greater dependency on IRE1 than its canonical target XBP1. Moreover, they show that while genetic silencing of IRE1 affects the growth of these myeloma cell lines, pharmacologic inhibition of its kinase or RNase activities does not. Hence, this manuscript attempts to explore and mechanistically understand the requirement of these myeloma cell lines for the nonenzymatic function of IRE1.

By examining data from the DepMap.Org Portal and their own transcriptomics results, they focused their examination on Interferon Regulatory Factor 4 (IRF4) as a possible downstream target of IRE1. IRF4 has been long known to be highly expressed in B cells and plasma cells and plays essential roles in controlling B cell to plasma cell differentiation and immunoglobulin class switching. IRF4 overexpression is often found in myeloma cells and required for their survival. This makes it an interesting topic of study here, but also a challenging one because it plays a critical role in myeloma cell survival and manipulating it likely has effects independent of IRE1.

They author show here that shRNA silencing of IRE1 but not XBP1 significantly downregulated the IRF4 mRNA and protein levels in AMO1, KMS27, and L363 cells. They fail to test what happens to IRF4 in myeloma cell lines that are dependent on IRE1 enzymatic activity and XBP1 expression, which should be done for comparison.

Nascent-mRNA analysis showed that IRE1 silencing significantly reduced de novo IRF4 transcription, but they were not able to connect this with any known IRF4-regulating transcription factors. Since IRF4 is an auto-stimulatory transcription factor, they explored the possibility that IRE1 regulates IRF4's activity itself. They performed subcellular fractionation and found that IRE1 silencing markedly decreased the relative amount of IRF4 bound to chromatin, suggesting a functional disruption of IRF4 activity. However, this is difficult experiment to quantify in this way and ChIP would be a better method.

The authors then present evidence that IRE1 silencing significantly increased IRF4 phosphorylation on Ser114 as well as Ser270, while pharmacologic IRE1 inhibition did not induce a significant increase. However, the quantification methods used are not well explained and make it difficult to assess confidence in the results. I am not sure if any phospho antibodies exists against these sites, but the authors do not assess them by immunoblotting.

A functional consequence of IRF4 phosphorylation on Ser114 or Ser270 does not seem to have been previously reported in the literature (although p38a MAPK has been reported capable of phosphorylating Ser270--PMID: 36443297). Therefore, in an attempt to examine the functional importance of these phosphorylation sites, the authors generated an IRF4 mutant in which these serine residues were replaced with alanine (S114A_S270A). Upon ectopic expression in AMO1 cells, this mutant displayed a 4-fold increase in its chromatin-binding capacity as compared to WT IRF4, indicating that phosphorylation on these sites inhibits IRF4 activity. However, as the ectopically expressed proteins are run on separate gels (Fig S2D), we are unable to assess whether their expression is comparable. Moreover, as they did not carry out transcriptional studies with these mutants we have no direct evidence on how these mutations affect their transcriptional activity. Moreover, they did not compare how IRF4 wt vs non-phosphorylatable mutants are able to rescue IRE1 depletion in myeloma cells. If their model is correct, IRE1 depletion should have negligible effects on cells expressing the IRF4 non-phosphorylatable mutant--an important experiment that was not done.

The authors then carry out a series of studies comparing IRF4 knockdown to IRE1 knockdown, find that both impair myeloma cell growth, and try to build a case that they are in linear pathway. Unfortunately, these experiments are not convincing given that loss of two parallel but important growth pathways can give similar effects on cell growth. These experiments are not particularly helpful or enlightening.

In Figure 6, the authors finally carry out their first IRF4 repletion studies to test if this rescues cells deficient in IRE1. IRF4 repletion during IRE1 silencing was sufficient to rescue proliferation of AMO1 cells (Fig 6A-B). Although IRF4 re-expression effectively restored growth for up to 72 h, a modest growth decline of up to 15% occurred beyond this time point. As mentioned, these are difficult experiments to interpret because it is possible that IRF4 expression may very well promote cell growth (as it is known to do), but does not mean that it is downstream of IRE1. Moreover, this is the experiment where the IRF4 non-phosphorylatable mutant should also be tested. Furthermore, IRF4 repletion only partially prevented apoptosis induction during IRE1 silencing, as measured by DNA fragmentation (Fig 6C) or caspase-3/7 activation, against suggesting that it is not sufficient.

In the end, the reader is left with more questions than answers. While the data suggest that a subset of myeloma cell lines require some nonenzymatic function of IRE1, the effects on IRF4 loss are largely phenomenological at this point. it is unclear if they are in a linear pathway. It is unclear if/how IRE1 regulates IRF4 mRNA levels and/or phosphorylation.

As such, this manuscript does not provide a substantial enough advance to justify publication in PLoS Biology.

Rev. 4:

The paper by Oikonomidi and colleagues examines the basis for a non-enzymatic dependency on IRE1 in a subset of MM cell lines and suggest that the dependency is mediated by the transcription factor IRF4. Here the authors begin by identifying a set of genes that are essential for the survival of IRE1-dependent MM cell lines and filtered this list for those that are transcriptionally dependent on IRE1 for expression. Through this approach, the authors identified IRF4 (among others) as a key target that was downregulated upon IRE1 silencing. From here, the authors then perform a large series of experiments to investigate the basis for the association between IRE1 and IRF4. The most novel of these is the finding that IRE1 silencing leads to the phosphorylation of IRF4, which disrupts the ability of IRF4 to engage DNA and drive transcription. The authors also show that IRF4 overexpression can rescue the anti-proliferative effect that is seen with the loss of IRE1.

Overall, the manuscript is well written, the experiments are well executed and extensive, and the findings, albeit in a select group of cell lines, will be of interest to the myeloma research community. Additionally, in the discussion, the authors acknowledge that further work is required to decipher just how IRE1 regulates IRF4 phosphorylation. There are a few questions however that this reviewer would like to see addressed. The most notable of these is the disconnect between how IRE1 is presumably only required for the survival for a small subset of MM cell lines, yet IRF4 is required for all.

Questions.

1. The authors should state how many MM cell lines (proportion of the ones they tested) they found to be IRE1 dependent. This is important as the Depmap data that they use extensively throughout the manuscript would argue that IRE1 loss has little effect on MM survival.

2. Additionally, the authors should indicate how many of the IRE1-dependent MM cell lines were independent of IRE1's enzymatic activity. This subset is likely even smaller, and this detail would add clarity to their findings.

3. In Figure 1, the authors exclude common essential genes from their analysis. They should provide a rationale for this decision. In Figure 2D, IRE1 silencing leads to the downregulation of the common essential gene MYC. Could MYC downregulation be driving the loss of cell viability independently of IRF4?

4. The authors also identified the key MM survival gene PRDM1 as an IRE1-dependent gene? The authors should provide a rationale as to why they focused on IRF4 and not PRDM1. It may be that they investigated PRDM1, and it led to a dead end.

5. The authors show that IRE1 silencing leads to the loss of both IRF4 transcript and protein in IRE1-dependent MM cell lines. What happens to IRF4 when IRE1 is silenced in MM cell lines that are NOT dependent on IRE1 for their survival. Including additional data here would significantly strengthen the paper and allow the authors to make the case that for this subset of MM cell lines, IRE1 is somehow uniquely important for IRF4 maintenance.

6. Finally, in the discussion, the authors should address the apparent discrepancy between IRE1's requirement in a subset of MM cell lines and IRF4's essentiality across all MM cell lines.

---

## [Decision Letter · Decision Letter 2]

28 Jan 2025

Dear Dr Ashkenazi,

Thank you for your patience while we considered your revised manuscript entitled "IRF4 Mediates Nonenzymatic IRE1 Dependency in Multiple Myeloma Cells" for publication as a Research Article at PLOS Biology. This revised version of your manuscript has been evaluated by the PLOS Biology editors, the Academic Editor and three of the original reviewers.

Based on the reviews, we are likely to accept this manuscript for publication, provided you satisfactorily address the data and other policy-related requests stated below.

In addition, we would like you to consider a suggestion to improve the title:

"Interferon regulatory factor 4 mediates non-enzymatic IRE1 dependency in multiple myeloma cells"

We expect to receive your revised manuscript within two weeks. 

*Published Peer Review History*

*Press*

Sincerely,

Ines

--

Ines Alvarez-Garcia, PhD

Senior Editor

PLOS Biology

DATA POLICY:

Fig. 1A-E, G-J; Fig. 2A-C, E-H; Fig. 3A-E; Fig. 4A, C, E; Fig. 5A, D, E; Fig. 6B-F, H-J; Fig. S1A, C, E, G, J, K; Fig. S2A, C; Fig. S3B, D, E, G-I; Fig. S4A-C and Fig. S6A-C, E, H-J

CODE POLICY

Reviewers' comments:

Rev. 2: Giovanni Tonon

I am satisfied with the answers to my points, minor corrections:

- figure 2J is not clear where it is, and where the results mentioned in the rebuttal letter are included

- alignment of top labels in S2C

Rev. 3:

The authors have adequately addressed all of my previous concerns with the addition of substantial new data. The manuscript is now much stronger and worthy of publication.

Rev. 4:

I am satisfied with the authors responses to my questions and feel that the manuscript is now acceptable for publication in your Journal.

---

## [Editor Report · Decision Letter 3]

4 Mar 2025

Dear Dr Ashkenazi,

Thank you for the submission of your revised Research Article entitled "Interferon Regulatory Factor 4 mediates nonenzymatic IRE1 dependency in multiple myeloma cells" for publication in PLOS Biology. On behalf of my colleagues and the Academic Editor, Ursula Jakob, I delighted to let you know that we can in principle accept your manuscript for publication, provided you address any remaining formatting and reporting issues. These will be detailed in an email you should receive within 2-3 business days from our colleagues in the journal operations team; no action is required from you until then. Please note that we will not be able to formally accept your manuscript and schedule it for publication until you have completed any requested changes.

PRESS

Sincerely, 

Ines

--

Ines Alvarez-Garcia, PhD

Senior Editor

PLOS Biology
